

# Application of a two-step approach for mapping ice thickness to various glacier types on Svalbard

Johannes Jakob Fürst[1], Fabien Gillet-Chaulet[2], Toby J. Benham[3], Julian A. Dowdeswell[3], Mariusz Grabiec[4], Francisco Navarro[5], Rickard Pettersson[6], Geir Moholdt[7], Christopher Nuth[8], Björn Sass[1], Kjetil Aas[8], Xavier Fettweis[9], Charlotte Lang[9], Thorsten Seehaus[1], and Matthias Braun[1]

[1]Institute of Geography, University of Erlangen-Nuremberg, Wetterkreuz 15, 91058 Erlangen, Germany
[2]University of Grenoble Alpes, CNRS, IRD, Institut des Géosciences de l'Environnement (IGE), CS 40 700, Grenoble, France
[3]Scott Polar Research Institute, University of Cambridge, Lensfield Road, Cambridge CB2 1ER, United Kingdom
[4]Faculty of Earth Sciences, University of Silesia in Katowice, ul. Bankowa 12, 40-007 Katowice, Poland
[5]Departamento de Matemática Aplicada a las Tecnologías de la Información y las Comunicaciones, desp. A302-4, ETSI de Telecomunicación. Universidad Politécnica de Madrid, Av. Complutense 30, 28040 Madrid, Spain
[6]Department of Earth Sciences, Uppsala University, Geocentrum, Villav. 16, 752 36 Uppsala, Sweden
[7]Norwegian Polar Institute, Fram Centre, Postbox 6606 Langnes, 9296 Tromsø, Norway
[8]Department of Geosciences, University of Oslo, Postboks 1047, Blindern, 0316 Oslo, Norway
[9]Department of Geography, University of Liège, Quartier Village 4, Clos mercator 3, 4000 Liège, Belgium

*Correspondence to:* Johannes Fürst (johannes.fuerst@fau.de)

**Abstract.** The basal topography is largely unknown beneath most glaciers and ice caps and many attempts have been made to estimate a thickness field from other more accessible information at the surface. Here, we present a two-step reconstruction approach for ice thickness that solves mass conservation over single or several connected drainage basins. The approach performs well for a variety of test geometries with abundant thickness measurements including marine- and land-terminating glaciers as well as a $2\,400\,\mathrm{km}^2$ ice cap on Svalbard. Input requirements for the first step are comparable to other approaches that have already been applied world-wide. In the first step, a geometrically controlled, non-local flux solution is converted into thickness values relying on the shallow ice approximation. In a second step, the thickness reconstruction is improved along fast-flowing glacier trunks on the basis of velocity observations. In both steps, thickness measurements are assimilated as internal boundary conditions. Each thickness field is presented together with a map of error estimates which stem from a formal propagation of input uncertainties. These estimates point out that the thickness field is least constrained near ice divides or in other stagnant areas. The error-estimate map also highlights key regions for future thickness surveys as well as a preference for across-flow acquisition. Withholding parts of the thickness measurements indicates that error estimates show a tendency to overestimate actual mismatch values. For very sparse or non-existent thickness information, our reconstruction approach indicates that we have to accept an average uncertainty of at least 25% in the reconstructed thickness field. For Vestfonna, previous ice volume estimates have to be corrected upward by 22%. We also find that a 12% area fraction of the ice cap are in fact grounded below sea-level as compared to the previous 5%-estimate.



# 1 Introduction

For the 210'000 glaciers and ice caps that we find on this planet (Bishop et al., 2004), satellite remote sensing based on optical or radar instruments has recently enabled us to measure changes in glacier extent and in surface elevation with virtually complete coverage (e.g. Zwally et al., 2011; Rankl et al., 2014; Paul et al., 2015; Zwally et al., 2015). Yet, for the large majority

of these ice geometries, there is no information available on the thickness of the ice cover (Gärtner-Roer et al., 2016). Any attempt to predict the glacier demise under climatic warming and estimate the future contribution to sea-level rise (Radić and Hock, 2011; Radić et al., 2014; Marzeion et al., 2012, 2014; Huss and Hock, 2015) is limited as long as the glacier thickness is not well known. Moreover, the ignorance of the bed topography inhibits the applicability of ice-flow models, which could help to understand dominant processes controlling the ice-front evolution of marine-terminating glaciers. This is because the basal

topography exerts a major control on the dynamic response of grounded ice (Schoof, 2007, 2010; Favier et al., 2014). A reason for further concern is that grounded parts of the Antarctica Ice Sheet are assumed to respond to climatic warming primarily by outlet glacier acceleration, as the floating ice-shelves thin (Paolo et al., 2015) and loose their buttressing ability (Fürst et al., 2016). As it is impractical to measure ice thicknesses everywhere, reconstruction approaches have been proposed that can infer thickness fields from available geometric, climatic and ice-velocity information.

In terms of input requirements, reconstruction approaches always need information on the geometric setting. This normally comprises the glacier outline and the surface topography. In the "Ice Thickness Models Intercomparison eXperiment" (ITMIX; Farinotti et al., 2016), two types of reconstruction approaches rely exclusively on this geometric information. The first type stems from a perfect plasticity assumption, relating ice thicknesses to a glacier-specific yields stress, which itself is inferred from the elevation range of the glacier (Linsbauer et al., 2012; Frey et al., 2014; Carrivick et al., 2016). The second type

assumes that characteristics of the ice-covered bed topography resemble the nearby ice-free landscape (Clarke et al., 2009). Under this premise, an artificial neural network is trained with digital elevation models (DEM) of the surrounding area. Once sufficiently trained, this approach can efficiently compute glacier bed topographies. Another reconstruction approach (Gantayat et al., 2014) uses additional information on surface velocities and relies on the shallow ice approximation (SIA; Hutter, 1983; Morland, 1986). Under this assumption, surface velocities directly translate into ice-thickness values dependent on

glacier-surface slopes. Most of the participating approaches rely, however, on mass conservation. This implies that they need information on the difference between the actual surface mass balance (SMB) and the contemporaneous surface elevation changes. Although this difference is often referred to as "apparent mass balance" (Farinotti et al., 2009b), we apply the term "apparent flux divergence" here. A large subset of these approaches generates a generic apparent flux divergence field informed by the geographic location and the continental character of the prevailing climate assuming specific linear relations to the

surface elevation below and above a preset equilibrium line altitude (Farinotti et al., 2009a; Huss and Farinotti, 2012; Clarke et al., 2013). In addition, these approaches rely on the SIA and require an input ice-discharge value for marine-terminating glaciers. As standard procedure, many of the above approaches dissect glacier outlines into a number of centrelines along which the actual reconstruction is performed. Consequently, these approaches are computationally efficient but they require a final interpolation of the thickness values between centrelines. To avoid such an interpolation, computationally less favorable mass





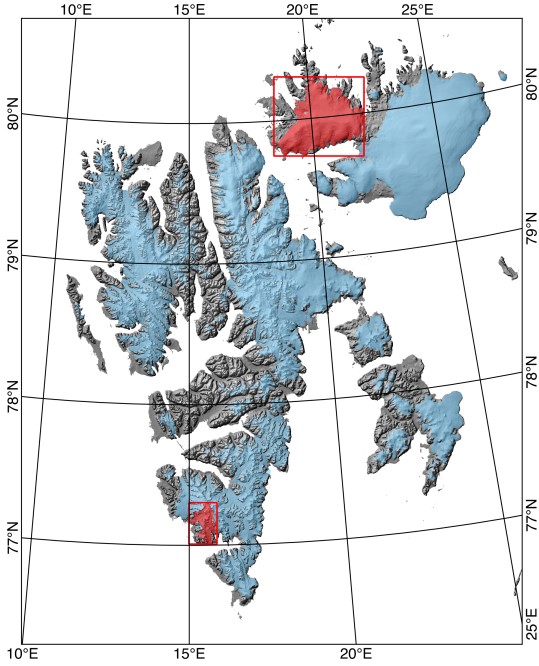

**Figure 1.** Overview map of the Svalbard archipelago showing ice coverage (blue shading). The two test sites (red shading and rectangles) are Vestfonna Ice Cap (VIC) on Nordaustlandet and the glacier complex comprising the marine-terminating Austre Torell-, Hans-, Paierlbreen (THPB) and the land-terminating Werenskioldbreen (WSB) in Wedel Jarlsberg Land. Background: grey-scale hill-shaded topography based on a 50m DEM from the Norwegian Polar Institute (NPI).

conservation approaches were adopted that find a physical solution over entire glacier basins (Morlighem et al., 2011; McNabb et al., 2012; Brinkerhoff et al., 2016). Two strategies are pursued for these reconstruction types. For the one type, ice-flow models are applied in a pseudo-transient way such that the actual surface elevation remains close to observations optimising the bed topography (van Pelt et al., 2013). Dependent on the dynamic complexity, these forward modelling approaches become

5   very expensive in terms of computing resources. For the other type, ice velocities are taken from observations and enter the mass conservation equation, which is then directly solved for ice thickness (Morlighem et al., 2011; McNabb et al., 2012; Mosbeux et al., 2016). Thickness measurements are more or less well accounted for in all of the above approaches. From an observational perspective, operational and regular satellite imagery acquisition has become an indispensable and continuously growing source of information. Therefore, automated procedures have been brought in place providing products such as glacier

10   outlines (Bishop et al., 2004; Atwood et al., 2010; Nuth et al., 2013; Rankl et al., 2014; Paul et al., 2015), digital elevation models (ArcticDEM; ASTER GDEM2, Tachikawa et al. (2011); SRTM, Farr et al. (2007),TanDEM-X, Rankl and Braun (2016)) and surface velocities (Joughin et al., 2010; Rignot et al., 2011; Rignot and Mouginot, 2012; Rosenau et al., 2015; Seehaus et al., 2015; Fahnestock et al., 2016; Seehaus et al., 2016). DEM differencing then allows to estimate surface elevation changes Much development effort is now put into reducing associated uncertainties in these measurements. Glacier outlines



are accessible globally and regular updates can be generated with existing methods. For DEMs and elevation changes, different data sources and methods exist. Depending on the missions, the products can be generated almost operationally for large areas. Surface velocities are somewhat more challenging to infer, certainly where magnitudes are small or where the glacier surface is featureless, such as in the accumulation area, or during periods of rapid changes in surface characteristics (for instance when

first melt or snowfall sets in). Consecutive image pairs can then de-correlate leaving gaps in the measurements. In addition, measurement uncertainties become relatively more important where velocities are small. In terms of mass conservation, spatial velocity gradients are required which are susceptible to small measurement uncertainties. The SMB field is another prerequisite for mass conservation. It is not directly measurable by remote sensing techniques. Sparse SMB records can be used to determine elevation gradients that are then extrapolated according to a regional DEM (Farinotti et al., 2009b). Otherwise, SMB records

are exploited to validate parametric SMB approaches (Möller et al., 2016) or more physically-based, low-resolution regional climate models could be applied (Lang et al., 2015; Aas et al., 2016).

In light of this continuously growing body of information, input fields for relatively complex thickness-reconstruction approaches are readily available. In this regard, we present a two-step approach that provides, in the first step, a physically based thickness field over entire glacier basins, ice fields or ice caps with few input requirements (Sect. 3.2). In the second step,

velocity information is exploited to update and improve the thickness reconstruction in specific areas (Sect. 3.3). A final interpolation of the basal topography is not required. For a set of three test geometries on Svalbard, the necessary input data were gathered (Sect. 2) and thickness maps were inferred. A rich thickness record is available on these test glaciers and serves to constrain both the ice-thickness distribution and the associated map of error estimates (Sect. 4).

## 2 Test geometries

The two-step thickness reconstruction approach is tested on three ice geometries on Svalbard where an abundant record of thickness observations was available (Fig. 1). These three test geometries comprise Vestfonna ice cap (VIC) on Nordaustlandet, the glacier complex composed of the marine-terminating Austre Torellbreen, Hansbreen, Paierlbreen (THPB) in Wedel Jarlsberg Land and in the same region the land-terminating Werenskioldbreen (WSB). This choice allows an assessment of how the approach performs under different glacier geometries. The reconstruction approach requires as input the glacier outline, the

surface geometry, the surface mass balance as well as surface elevation changes. Optionally, ice velocity information can be used to improve the thickness reconstruction in a second step. Fjord bathymetry information and thickness measurements are used to constrain the inferred thickness values.

### 2.1 General characteristics

VIC is one of the two major ice caps on Nordaustlandet (Dowdeswell, 1986a), being the largest in the Svalbard archipelago

(Fig. 1). According to the 2002-2010 glacier inventory, it covers an area of 2366 km$^2$ with its summit area lying at 630m above sea level (a.s.l.). Ice flow is channeled through several elongate outlet glaciers, which drain radially from a central crest and export ice to the surrounding ocean (Fig. 2g). For the outlet glaciers on the south side of Vestfonna (Dowdeswell and Collin,





1990), ice front positions have retreated in the period 1984-2010 (Braun et al., 2011) after more variable terminus fluctuations in the 1970s (Dowdeswell, 1986b). Despite the steady retreat for most outlet glaciers, Søre and especially Nordre Franklinbreen on the west side advanced notably. This re-advance coincided with a strong acceleration reaching far inland. Surface velocities doubled and now exceed 100 m yr$^{-1}$ over a large area (Pohjola et al., 2011). Prior to the speed-up, most of the ice was exported

via the northern branch of Nordre Franklinbreen. In the meantime, ice velocities indicate that the southern branch is the more prolific export path (Fig. 2g). The bi-modal pattern in ice dynamics is overprinted by cyclic surges with the last active phase observed in 1952 along Søre Franklinbreen (Błaszczyk et al., 2009). Surges are quasi-periodic cycles of an active phase, during which extremely fast flow can transfer an immense ice volume downglacier, followed by a quiescent phase during which the ice cover in the accumulation area gradually regains its former height. Two other surge-type glaciers are known in the eastern part

of Vestfonna. Active phases were reported in 1939 and 1992 for Rijpbreen and during the period 1973-1980 for Bodleybreen (Dowdeswell, 1986b; Błaszczyk et al., 2009).

Austre Torellbreen is a marine-terminating glacier in Wedel Jarlsberg Land, southern Spitsbergen (Fig. 2b). The glacier drains into Skoddebukta on Hornsundbanken and spans altitudes from sea-level to about 900m a.s.l. The most elevated parts of the accumulation area belong to Amundsenisen (above 700m a.s.l.) and are drained by Bøygisen and Løveisen. Before reaching

the ocean, Austre Torellbreen is further fed by Vrangpeisbreen from the south, which shares an ice divide with Hansbreen. Hansbreen, in turn, has a dominant main branch receiving important lateral inflow from two prominent tributaries in the southwest, i.e. Deileggbreen and Tuvbreen (Grabiec et al., 2012). The glacier shows a somewhat reduced elevation range of only up to 500m a.s.l. Beyond the mountain range to the east lies Paierlbreen covering again the full elevation range. This glacier also connects to Amundsenisen in the north via Nornebreen. Kvitungisen provides a large passage to Hansbreen in the west.

Lågberisen reaches up to the central divide areas of the glacier complex whereas Perlebreen joins further downstream from the eastern side, just before the ocean is reached. Paierlbreen was not only classified as marine-terminating in the 2002-2010 glacier inventory but the glacier also exhibited surge behaviour in 1993-1999 Błaszczyk et al. (2009); Nuth et al. (2013). During the surge, the ice front position was, however, not much affected. The reason might be that the surge event was superimposed on the well-documented retreat of all marine-terminating glaciers in the Hornsund area over the last century (Błaszczyk et al.,

2013). Austre Torellbreen, Hansbreen and Paierlbreen cover areas of 141, 64 and 99 km$^2$, respectively. West of the THPB complex lies Werenskioldbreen (Ignatiuk et al., 2014). It is land-terminating and somewhat smaller with 27 km$^2$. It covers an elevation range from 40 to just above 700m a.s.l.

## 2.2 Glacier outlines

Glacier outline information is taken from the 2002-2010 glacier inventory described in Nuth et al. (2013). As THPB is a

well-connected glacier complex, adjacent glacier boundaries were removed and joined into one single outline. WSB was not merged with the THPB complex because the shared ice divide is short and shallow (Kosibapasset has only $\sim$ 15m depth). VIC is treated as a single entity by merging all its individual drainage basins. In this way, we avoid discontinuities in the anticipated thickness solution across ice ridges and divides.



## 2.3 Surface elevation

Concerning the Svalbard surface elevation, we rely on a 50m digital elevation model (DEM) from the 1990s[1] provided by the Norwegian Polar Institute (NPI). This map was produced from areal photos using photogrammetry as well as from contour lines in earlier elevation maps, which were digitised and interpolated. We refrained from using this DEM on VIC because it mainly

stems from contour-line information resulting in a characteristic wave pattern in the slope field. Therefore, we use a more recent 10m DEM inferred from 2010 radar data acquired by the TanDEM-X mission, operated by the German Aerospace Center (DLR; Krieger et al., 2013). The DEM was processed from bi-static Synthetic Aperture Radar (SAR) data using a differential interferometric approach (Seehaus et al., 2015; Rankl and Braun, 2016; Vijay and Braun, 2016, ; Saß et al., (in preparation)). It was referenced to sea level by means of ICESat (Schutz et al., 2005).

## 2.4 Thickness measurements

For VIC, thickness measurements (Fig. 4a) were obtained from 60MHz airborne radio-echo sounding surveys between 1983 and 1986 (Dowdeswell et al., 1986). Five flightlines run north-south across the ice cap and two from east to west. All profiles follow centrelines of prominent outlet glaciers. Unfortunately, no bed reflector could be identified for a large portion of these airborne data, including most of the ice-divide area. There only recently (2008-2009), ground-based pulsed radar data

were collected by (Pettersson et al., 2011). Following Pettersson et al. (2011), the early airborne measurements were adjusted assuming a constant thinning rate of $\sim 0.16$m yr$^{-1}$ over the entire ice cap.

    In the Hornsund area, Hansbreen is well studied and an ice-core drilling team reached the bed at three locations already in 1994 (Jania et al., 1996). Between 2004 and 2013, ground-penetrating radar profiles were collected both on THPB and WSB (Navarro et al., 2014). These surveys provide a dense grid over most parts of these glaciers (Fig. 4b). For WSB, the

measurement error was analysed in depth even accounting for the often ignored positioning-related ice-thickness uncertainty (Lapazaran et al., 2016). Measurement errors fall into a range of 3.3 to 6.8 m with an average value of 4.5 m. The early ice-core information was ignored for the thickness reconstruction here because it only gives information at a few additional points and because it is not evident how the surface elevation has changed since the early 1990s.

## 2.5 Surface mass balance

For the SMB information, we rely on the regional climate model MAR (Modèle Atmopshérique Régional; Lang et al., 2015). MAR combines a hydrostatic model for the atmospheric circulation with a physically based model for snow-pack evolution. The MAR-SMB simulations cover the entire archipelago (Fig. 2a,b) and were validated by Lang et al. (2015) against available climatic variables as well as SMB measurements from Pinglot et al. (1999, 2001). Simulation were conducted on a regular 7.5km grid but a downscaled output was provided on 200m spacing using an interpolation strategy that distinguishes the various

SMB components (Franco et al., 2012). The components are interpolated according to locally defined, vertical gradients. For the reconstruction, the annual SMB record was averaged over 1979-2015.

---

[1]Norwegian Polar Institute (2014). Terrengmodell Svalbard (S0 Terrengmodell) [Data set]. Norwegian Polar Institute. doi:10.21334/npolar.2014.dce53a47





**Figure 2.** Input fields to the ice thickness reconstruction for VIC (a,c,e,g) and THPB/WSB (b,d,f,h). Surface mass balance (SMB) input (a,b) is provided by MAR as an average over the period 1979-2014 (Lang et al., 2015). Elevations changes (c,d) are inferred from 2003-2007 ICESat profiles on VIC and from a 2008 SPOT-HRS DEM in southern Spitsbergen. From this elevation information, we subtracted the 1990 DEM from the Norwegian Polar Institute (NPI). For VIC, line information on elevation changes along the ICESat tracks was linearly interpolated (c). The difference between SMB and surface elevation changes (e,f) is referred to as the apparent flux divergence. Surface velocity magnitudes (g,h) were inferred from 2015/2016 Sentinel-1 imagery. Background: grey-scale hill-shaded topography based on a NPI 50m DEM.





**Figure 2. (continued)**

To assess the sensitivity of the thickness reconstruction to the SMB input (Appendix B1), results from the Weather Research and Forecasting (WRF) model were considered (Aas et al., 2016). The WRF-SMB field represents the period 2003-2013 and has a 3km resolution. The field could not be downscaled as the above routines were not implemented by ourselves. Therefore, the SMB sensitivity is only assessed on the large VIC geometry.





## 2.6 Surface elevation changes

Over VIC, 2003-2007 elevation changes (Fig. 2c) were inferred from laser altimetry measurements with the Ice, Cloud, and Land Elevation Satellite (ICESat). ICESat measurements were referenced to the 1990 20m NPI DEM[1] (Moholdt et al., 2010). The laser altimetry system has a footprint of 70m diameter with 170m along-track spacing. Across-track spacing is irregular
and much larger with several kilometres. A Natural-Neighbour Sibsonian interpolation[2] (Fan et al., 2005) is used to estimate elevations changes in between these scattered ICESat measurements.

For Wedel Jarlsberg Land, elevation changes were calculated by differencing the NPI 20m DEM[1] from 1990 with a 40m DEM inferred from 2008 imagery acquired by the high resolution stereoscopic (HRS) sensor on-board SPOT 5 (Korona et al., 2009). The DEMs were first co-registered (Nuth and Kääb, 2011) before subsequent differencing and resampling to 100m
(Fig. 2d).

## 2.7 Surface velocities

Using satellite imagery acquired between January 2015 and September 2016 by the C-band Synthetic Aperture Radar (SAR) onboard Sentinel-1, we apply intensity offset tracking to consecutive image pairs (Strozzi et al., 2002; Seehaus et al., 2016). The time series of displacement fields is first filtered for obvious outliers within a certain kernel area in terms of the prevailing flow
direction and magnitude (Seehaus et al., 2016). Then, fields are stacked using median-averaging to obtain maximum coverage and to reduce effects from short-term or seasonal fluctuations (Fig. 2g, h). Velocity maps are provided at 100m resolution. The uncertainty associated to the inferred velocity maps is estimated on 70 stable reference areas without ice cover. We find an average uncertainty of 19 m yr$^{-1}$, which is comparable to independent uncertainty estimates for merged Sentinel-1 imagery with minimum values of $\sim 17$m yr$^{-1}$ (Schwaizer, 2016).

## 2.8 Fjord bathymetries

Information on the fjord bathymetry is used to further constrain the thickness reconstruction at marine ice fronts. The new International Bathymetric Chart of the Arctic Ocean (IBCAO Version 3.0) holds a wealth of new measurements around the Svalbard archipelago (Jakobsson et al., 2012). It comprises several recent multibeam surveys that entered deep into some major fjords and collected high-resolution seafloor information (Ottesen et al.). Around the archipelago, the new IBCAO map
is provided at high spatial resolution of 500m. To some extent, this fine spacing is possible because of high-resolution input data from the *Olex* seabed mapping system.

## 3 Methods

The thickness reconstruction approach is ultimately based on mass conservation and largely stems from ideas presented in Morlighem et al. (2011). We opted for a two-step approach because surface velocity information from satellite remote sensing

---

[2]source code available at https://github.com/sakov/nn-c/tree/master/nn





often fails to cover an entire glacier basin. The two-step approach, first provides a thickness field over the entire glacier surface which can be updated in areas where reliable velocity information is available. In the first step, a balance flux is calculated from the difference between SMB and surface elevation changes, i.e. the apparent flux divergence. Relying on the SIA (Hutter, 1983), the local ice thickness is calculated from the resultant ice-flux field and the surface topography. In a second step, the

mass conservation equation is directly solved for ice thickness in a sub-domain where reliable velocity information is available. The first step thickness field then serves as lateral boundary condition.

### 3.1    Mass conservation

Assuming that ice is an incompressible material, the 3D velocity field $\boldsymbol{v}$ is free of divergent or convergent flow. Any divergence in the vertically-averaged, horizontal velocity components $\boldsymbol{u} = (u_1, u_2)$ translates directly into thickening or thinning. Ac-

counting for the kinematics of the upper and lower boundary surfaces (p. 333 in Cuffey and Paterson, 2010), incompressibility can be rewritten as follows.

$$\frac{\partial H}{\partial t} + \boldsymbol{\nabla}\left(\boldsymbol{u}H\right) = \dot{b}_{\mathrm{s}} + \dot{b}_{\mathrm{b}} \tag{1}$$

Here, $\Omega$ is the 2D ice-covered domain. The flux divergence term is a-priori unknown and we rearrange accordingly.

$$\boldsymbol{\nabla}\boldsymbol{F} = \dot{a} \tag{2}$$

Here, $\nabla$ is the divergence operator in two dimensions, $H$ is the ice thickness and $\boldsymbol{F} = \boldsymbol{u}H$ is the horizontal ice-flux vector. Sources and sinks in the flux divergence arise from the surface $\dot{b}_{\mathrm{s}}$ and bottom mass balance $\dot{b}_{\mathrm{b}}$, as well as from surface elevation changes $\partial H/\partial t$ over time. We avoid the terminology of apparent mass balance for the combined effect of all source and sink terms $\dot{a} = \dot{b}_{\mathrm{s}} + \dot{b}_{\mathrm{b}} - \partial H/\partial t$ and rather refer to it as apparent flux divergence (Fig. 2e, f). Except for glacier geometries that are in equilibrium with the climatic conditions ($\partial H/\partial t = 0$), $\dot{a}$ is not directly associated with the mass balance neither for a glacier

as a whole nor at the surface. Throughout this manuscript, we assume that the basal mass balance $\dot{b}_{\mathrm{b}}$ is negligible as compared with the other terms in the apparent flux divergence.

### 3.2    First step: Flux-based solution

In a first step, the mass conservation is solved for the ice flux (Eq. 2), which itself is assumed to follow the steepest descent in the surface topography.

$$\tilde{\boldsymbol{F}} = \tilde{F} \cdot \boldsymbol{n} \tag{3}$$

Here, $\boldsymbol{n}$ is the normalised negative surface slope vector. Along all land-terminating segments of the glacier outline, we impose a zero-flux condition. A free boundary condition is chosen across marine ice fronts, providing an ice-discharge estimate consistent with the apparent flux divergence. Inflow boundaries are avoided because a Dirichlet condition on the ice flux would become necessary there. To solve Eq. (2), we use the Elmer finite-element software developed at the Center for Science in

Finland (CSC-IT, http://www.csc.fi/elmer/) and more specifically the mass conservation solver implemented in its glaciological





extension Elmer/Ice (Gagliardini and Zwinger, 2008; Gillet-Chaulet et al., 2012; Gagliardini et al., 2013). For the discretisation of the problem, we select the stabilised streamline upwind Petrov-Galerkin (SUPG) scheme (Brooks and Hughes, 1982).

The assumption that the ice-flux points into the opposite direction of surface gradients has important consequences for the solution to Eq. (2). The flux solution becomes highly dependent on the surface elevation information. Independent of the quality

or the resolution of the input DEM, singular source or sink points are introduced over the entire glacier domain where the DEM shows local minima or maxima. At these locations, flowlines have to either start or end resulting in a highly partitioned and erratic flux field. Even a topography smoothing using various kernel sizes could not suppress this effect. Searching for an effective way to avoid such singular points, we decided to first extract surface elevation contours each 50m intervals. This contour information is then interpolated and gridded using a Natural Neighbours Sibsonian interpolation (Fan et al., 2005).

This decision largely guarantees that the flux solution changes gradually and is free of most singular points.

In areas near the ice divide or near marine ice fronts, surface slopes can locally become very small. As a consequence, the ice-flux solution can diverge. Therefore, we decided to introduce a slope threshold $\alpha_0$ of $1°$. If this value is not reached, the normalisation of the slope vector uses this threshold. The chosen threshold value is small as compared to the two-fold values chosen in the original thickness reconstruction approach by Farinotti et al. (2009b). Both their thresholds lie higher at $5°$ and

$20°$. For VIC, THPB and WSB, the DEM surface slopes do not reach the $5°$-threshold over most of the respective glaciated area (97.2, 86.3 and 53.1%, respectively).The $1°$-threshold slope implies that this fraction reduces to 18.7, 4.8 and 0.1% of their respective surfaces and that this area disintegrates into many small patches distributed over the domain. Where the slope magnitude falls below the $\alpha_0$-threshold, the ice-flux direction is taken from the spatial gradient in the input SMB. The choice for a smaller threshold value, here, is in agreement with a thickness reconstruction on the Patagonian ice fields (Carrivick et al.,

2016), for which a value of $1.7°$ was applied.

### 3.2.1   Cost function and single-variate optimisation

The direct flux solution to all input fields often shows wide-spread negative values and high spatial variability. Therefore, we chose to iteratively update the apparent flux divergence $\dot{a}$, as control variable, such that undesired characteristics in the flux field are reduced. We anticipate that the ice-flux direction is positive and that it is smooth. For the purpose of the iterative

optimisation, we introduce the following cost function $J$.

$$J = \lambda_{\mathrm{pos}} \cdot \int_{\Omega} F^2 \int_{-\infty}^{F} \delta(s) ds d\Omega + \lambda_{\mathrm{reg}} \cdot \int_{\Omega} \left(\frac{\partial F}{\partial x}\right)^2 + \left(\frac{\partial F}{\partial y}\right)^2 d\Omega + \lambda_{\dot{a}} \cdot \int_{\Omega} \left(\dot{a} - \dot{a}^{\mathrm{init}}\right)^2 d\Omega \qquad (4)$$

Here, $\delta(s)$ is the Dirac delta function with $s \in \mathbb{R}$. The first term is zero for positive flux values but penalises negative flux solutions. The second term is a regularisation which favours smooth flux solutions. The last term increases the more the iteratively updated apparent flux divergence $\dot{a}$ deviates from the initial input. The cost $J$ should primarily be considered as a

function of $\dot{a}$. As the control variable is iteratively updated, the cost should decrease. Good performance was achieved with the following multiplier choice: $\lambda_{\mathrm{pos}} = 10^2$, $\lambda_{\mathrm{reg}} = 10^0$ and $\lambda_{\dot{a}} = 10^{-2}$. The parameter choice aimed at a balance between





improving the smoothness of the flux field and reducing areas with negative flux values by adapting $\lambda_{\text{pos}}$ and $\lambda_{\text{reg}}$. The solution showed not much sensitivity to changes in $\lambda_{\dot{a}}$.

For the optimisation of the cost function, we rely on the "m1qn3" module (Gilbert and Lemaréchal, 1989) that can solve large-scale unconstrained minimisation problems. It requires first derivatives of the cost with respect to the single control
variables $\dot{a}$. For a precise calculation of these derivatives, we rely on the adjoint system associated to Eq. 2. The stopping criterion for the iterative optimisation is non-dimensional at $10^{-14}$ and computed as a ratio between the current and the initial norm of the cost derivatives.

### 3.2.2 Inferring ice thickness

Once a flux field is determined over the glacier domain, the ice thickness is inferred in a post-processing step. Having assumed
that the ice flux follows the surface topography, a natural choice for relating it to ice thickness is the SIA (Hutter, 1983). This ice-dynamic approximation is typically applied to geometries with large aspect ratios (horizontal vs. vertical scales). It implies that the ice-flux magnitude is given by the local geometry.

$$F^{\star} = \frac{2}{n+2} B^{-n} \left(\rho g\right)^{n} \left\|\boldsymbol{\nabla} h\right\|^{n} \cdot H^{n+2} \tag{5}$$

Other parameters needing specification include the ice density $\rho = 917\,\mathrm{kg\ m^{-3}}$, the gravitational acceleration $g = 9.18\,\mathrm{m\ s^{-2}}$
and the flow law exponent $n = 3$. In this way, the ice thickness becomes highly dependent on gradients in surface elevation $\boldsymbol{\nabla} h$. Note that this assumption implies zero basal sliding and thus limits the applicability to areas of slow ice flow. This constraint will be reduced during the second step of this reconstruction (Sect. 3.3), which is informed by ice velocity measurements. The ice-viscosity parameter $B$ is a-priori unknown. Yet where thickness measurements are available, $B$ can be computed from Eq. (5). Thereafter, the scattered information on the ice-viscosity parameter $B$, at the thickness measurement locations,
is interpolated over the entire glacier domain. To avoid unreliable extrapolation effects, we prescribe a mean value for the viscosity parameter $B$ around the lateral domain margin. Per domain, the mean value is the average viscosity inferred at all measurement locations. If no thickness measurements had been available, an a-priori choice of the viscosity parameter would have been required.

Here, surface slopes are computed from a somewhat smoothed variant of the DEM. First, the original topographic map on
50m resolution (12 m for VIC) is downsampled to 100m (500m for VIC and THPB). Thereafter, a least-square difference parabola is fitted to each point, considering neighbouring DEM points within 300m (1500m for THPB and VIC). Local slopes are then calculated from the parabolic fit. This smoothing is an attempt to account for the SIA assumption on small aspect ratios.

In Eq. (5), the flux solution does not enter directly but is corrected to avoid negative thickness values. The reason is that
despite the cost term on negative ice flux, negative values persist in some small areas. On VIC and THPB, the area fraction with negative ice flux is 0.5 and 1.7%, respectively. On WSB however, the flux solution over the main branch is generally very small and shows many zero transitions. Consequently, the area-fraction is higher at 4.4%. The reason is that the apparent flux divergence shows no dominant source area in the upper glacier ranges. The zero transitions in the flux solution would directly





transmit into the ice thickness field. To avoid these oscillations, we correct the flux as follows: $F^\star = (1-\kappa)\cdot\|F\|+\kappa\cdot F_{\text{crit}}$ with $\kappa = 1-2/\pi\cdot\text{atan}(F^2/F_{\text{crit}}^2)$. $F_{\text{crit}}$ is equal 10% of the average flux magnitude over the domain. Along the lateral land-terminating domain margin, we keep $F = F^\star = 0$. In the case of abundant thickness measurements, the effect of this flux correction on the inferred thickness is compensated by the ice-viscosity choice. If no observations are available and for $F > F_{\text{crit}}$, the functional

dependence implies that the reduction effect on the inferred thickness field remains below 2% as compared to the uncorrected case. Where the ice flux exceed the domain average ($10 \cdot F_{\text{crit}}$), the effect on the ice thickness falls below 0.15%. Below $F_{\text{crit}}$, thickness values are effectively increased. The sensitivity of the results to this flux correction is assessed in Appendix B3.

### 3.2.3   Formal error estimate

Together with the thickness map, we want to present a formal error map. For this purpose, the uncertainty on the input fields, i.e.

the SMB and $\partial H/\partial t$, are propagated in two steps. Uncertainties are first transmitted through the mass conservation equation (Eq. 2) and the resulting estimate of the flux error is then scaled by a SIA flux-thickness conversion (Eq. 5). For the first step, we follow the ideas presented in Morlighem et al. (2014), who assume that the inaccurate flux field $F + \delta F$ also satisfies mass conservation.

$$\boldsymbol{\nabla}\left[(F + \delta F)\cdot(\boldsymbol{n} + \delta\boldsymbol{n})\right] = \dot{a} + \delta\dot{a} \tag{6}$$

Here, $\delta\boldsymbol{n}$ is the error on the prescribed flux direction. Neglecting second order terms and accounting for the fact that $F$ satisfies Eq. (2), the flux error is a solution of:

$$\boldsymbol{\nabla}\left[\boldsymbol{n}\delta F\right] = \delta\dot{a} - \boldsymbol{\nabla}\left[F\delta\boldsymbol{n}\right] \tag{7}$$

At the thickness measurement locations, we assume that the ice flux is known with a precision that is equivalent to the uncertainty in the thickness measurements $\delta H_{\text{obs}}$. For this purpose, the above reported 5m thickness-measurement uncertainty is

translated into a flux-equivalent value using Eq. (5) without correction $F^\star = F$. Along the land-terminating domain margin, we assume zero flux and the thickness error estimate implicitly becomes zero. Despite boundary conditions, the flux error estimate is actually constrained from measurements upstream and downstream. This is readily accounted for by solving the following two problems.

$$\boldsymbol{\nabla}\left[(+\boldsymbol{n})\,\delta F_1\right] = \delta\dot{a} + \|\boldsymbol{\nabla}\left[F\delta\boldsymbol{n}\right]\|$$
$$\boldsymbol{\nabla}\left[(-\boldsymbol{n})\,\delta F_2\right] = \delta\dot{a} + \|\boldsymbol{\nabla}\left[F\delta\boldsymbol{n}\right]\| \tag{8}$$

These two problems are structurally identical to Eq. (1) and thus numerically solved as described in Sect. 3.2. These two formal error estimates $\delta F_1, \delta F_2$ subsequently enter a linear error propagation within the SIA thickness-flux relation (Eq.5). This yields:

$$\delta H_i = \frac{1}{n+2}\left[-\frac{2}{n+2}B^{-1/n}\left(\rho g\right)^n\|\boldsymbol{\nabla}h\|^n\right]^{-1/(n+2)}\cdot\|F\|^{-(n+1)/(n+2)}\cdot\|\delta F_i\| \qquad i\in\{1,2\} \tag{9}$$

Finally, the thickness error estimate $\delta H$ is the minimum of the two values stemming from up- and downwards propagation

$\min(\delta H_1, \delta H_2)$.





Input fields to the calculation of this formal error map are the uncertainty associated with the thickness measurements (Lapazaran et al., 2016), the flux direction and the apparent flux divergence. These uncertainties are assumed to be constant: $\delta H_{\mathrm{obs}} = 5.0\,\mathrm{m}$, $\delta n = 0.2$ and $\delta \dot{a} = 0.2\mathrm{m}$ i.e. $\mathrm{yr}^{-1}$.

### 3.3 Second step: Velocity-based solution

In a second step, the ice thickness map is updated in areas where reliable surface velocity information is available by solving Eq. (1) directly for the ice thickness. Equation (1) is vertically integrated and surface velocity information needs to be translated into a vertical mean value. Within the scope of this methodological study, we apply this second step exclusively where velocity magnitudes exceed $100\,\mathrm{m}\,\mathrm{yr}^{-1}$ (details of the sub-domain delineation in Sect. 3.4). In these sub-domains, basal sliding is assumed to dominate over internal deformation, and therefore vertical mean and surface velocities are set equal. We rely on the same Elmer/Ice routine to discretise and solve the mass conservation problem as above (Sect. 3.2). Thickness measurements are imposed as internal Dirichlet boundary conditions, whereas the previously inferred thickness values are prescribed around the lateral domain margin. If the sub-domain comprises a marine ice front, no boundary conditions are imposed.

#### 3.3.1 Cost function & Multi-parameter optimisation

Again, the ice thickness solution is optimised as we cannot anticipate that input fields are consistent in terms of the mass balance equation. Yet in this step, the optimisation makes use of three control variables. The apparent flux divergence is complemented by the horizontal velocity components. For this second-step optimisation, a new and more elaborate cost function $N$ is defined.

$$N = \gamma_{\mathrm{pos}} \cdot \int_{\Omega} H^2 \int_{-\infty}^{H} \delta(s)ds\,d\Omega + \gamma_{\mathrm{marine}} \cdot \int_{\Gamma_{\mathrm{marine}}} H^2 \int_{-\infty}^{H} \delta(s-H_{\mathrm{min}})ds \int_{H}^{\infty} \delta(s-H_{\mathrm{max}})ds\,d\Gamma +$$

$$\gamma_{\mathrm{reg}} \cdot \int_{\Omega} \left(\frac{\partial H}{\partial x}\right)^2 + \left(\frac{\partial H}{\partial y}\right)^2 d\Omega + \gamma_{\dot{a}} \cdot \int_{\Omega} \left(\dot{a} - \dot{a}^{\mathrm{init}}\right)^2 d\Omega + \gamma_{\mathrm{U}} \cdot \sum_{i=1}^{2} \int_{\Omega} \left(u_{\mathrm{i}} - u_{\mathrm{i}}^{\mathrm{init}}\right)^2 d\Omega \qquad (10)$$

Most of the terms have equivalents in Eq. (4). As before, we penalise a negative solution, high variability of the control variables and the control-variable mismatch to initial values. The only new term is the line integral along any marine boundary $\Gamma_{\mathrm{marine}}$. It penalises thickness values outside a certain range. The lower limit of this range stems from the fact that marine-terminating glacier margins on Svalbard are mostly grounded (Dowdeswell, 1989). Therefore, $H_{\mathrm{min}}$ is given by the flotation criterion $H_{\mathrm{min}} = h \cdot \rho_{\mathrm{water}}/(\rho_{\mathrm{water}} - \rho_{\mathrm{ice}})$. The upper limit is calculated from the IBCAO bathymetry. We assume that the bed topography does not significantly decrease inland and thus that the bathymetry along the ice front should be shallower than the maximum depth at all ocean points within a 5-km radius. We mostly experienced that the lower limit is not reached. The multiplier choices are motivated as follows. The most decisive multiplier is $\gamma_{\mathrm{reg}}$ as it determines the smoothness. If chosen to high, boundary thickness values and measurements are simply smoothed without much consideration for the ice dynamic influence. If chosen too low, the thickness solution of adjacent flow lines decouples. The choice $\gamma_{\mathrm{reg}} = 10^{-2}$ represents a trade-off between the two extremes. Second, we deemed it appropriate to set $\gamma_{\mathrm{pos}} = \gamma_{\mathrm{marine}}$. This multiplier was gradually increased





until the solution was affected, suggesting $\gamma_{\mathrm{pos}} = 10^2$. As before, the remaining two multipliers $\gamma_{\dot{a}} = 10^{-4}$ and $\gamma_{\mathrm{U}} = 10^{-8}$ are not very decisive and they were mostly added to prevent general divergence.

As above (Sect. 3.2.1), cost derivatives with respect to the control variables $\dot{a}$ and $u_i$ were computed from the adjoint system to Eq. (2). Without further modifications, the iterative optimisation preferentially modifies $\dot{a}$ because the control variables have different magnitudes. To align relative change values, a scaling factor of 0.05 for the velocity derivatives was introduced. Convergence of this second-step optimisation is reached using the same threshold criterion as above (Sect. 3.2.1).

### 3.3.2  Error estimate

Errors are again estimated following the ideas presented in Morlighem et al. (2014). As the ice thickness is calculated directly from mass conservation, errors have only to be propagated through Eq. (1). By analogy with Sect. 3.2.3, two systems of equations limit the error estimate from upstream and downstream.

$$
\begin{aligned}
\boldsymbol{\nabla}\left[(+\boldsymbol{u})\,\delta H_1\right] &= \delta\dot{a} + \|\boldsymbol{\nabla}\left[\mathrm{F}\delta\boldsymbol{u}\right]\| \\
\boldsymbol{\nabla}\left[(-\boldsymbol{u})\,\delta H_2\right] &= \delta\dot{a} + \|\boldsymbol{\nabla}\left[\mathrm{F}\delta\boldsymbol{u}\right]\|
\end{aligned}
\tag{11}
$$

The minimum value of the absolute values of these two error estimates gives the actual thickness error $\delta H = \min(\|\delta H_1\|, \|\delta H_2\|)$. Input uncertainties are $\delta u = 20.0\,\mathrm{m\ yr^{-1}}$ and $\delta\dot{a} = 0.2\,\mathrm{m\ i.e.\ yr^{-1}}$

### 3.4  Gridding & Boundary conditions

The individual glacier outlines from Nuth et al. (2013) are first partitioned into marine and land-terminating segments by searching if surface elevation is zero within 150m of the outline point. For the THPB complex, the DEM showed more advanced glacier fronts than in the glacier inventory. For these glaciers, an acceptable choice for detecting marine termination was to use a 100m surface-elevation threshold instead. Subsequently, nunataks were automatically accounted for in the mesh, if resolved by the target grid spacing. The target mesh resolution was 200m for THPB and VIC and 100m for WSB. In addition, we added grid points at each location where thickness measurements were available. This was necessary to prescribe internal boundary conditions on ice thickness and error estimates. The observations are very densely spaced and we decided to only keep measurements that are more than 50m apart, which is half of the minimum grid spacing. The initial 20792, 44921 and 21273 measurements of VIC, THPB and WSB were thus reduced to 4475, 5945 and 1189 points, respectively. From the outline and measurement locations, a 2D mesh with triangular elements was generated using the open source finite element grid generator Gmsh (Geuzaine and Remacle, 2009). Nodal values for all input fields are determined relying on a standard Natural Neighbours Sibsonsian interpolation procedure (Fan et al., 2005).

In the first-step reconstruction, two external boundary conditions were necessary around the glacier domain. At outflow boundaries along marine ice fronts, no conditions were set on either the ice flux or the ice thickness. Also the shared ice divide between Austre and Vestre Torellbreen is an outflow boundary for Torellbreen. Where glaciers terminate on land, a zero flux Dirichlet condition was imposed. Internal boundary conditions were applied where measurements were available. There, flux error estimates $\delta\mathrm{F}_1, \delta\mathrm{F}_2$ were set in accordance to the reported 5m mean error on the thickness measurements



(Sect. 3.2.3). In the second step reconstruction, the domain is reduced to sub-domains with reliable velocity information. In each drainage basin, the largest sub-domain was chosen from all areas in which velocity observations exceed 100 m yr$^{-1}$. At the lateral boundaries of this sub-domain, ice-thickness values as well as thickness error estimates were prescribed from the first-step reconstruction. No boundary conditions were imposed along marine ice fronts. Where thickness measurements were

acquired, Dirichlet conditions were imposed on the thickness solution (Eq. 2) while values within the up- and downstream error propagation (Eq. 11) were set to the 5m measurement uncertainty (Sect. 3.2.3).

## 4   Results & Discussion

### 4.1   First step reconstruction

This section covers the presentation and discussion of the ice-flux solution, the reconstructed thickness and bedrock elevation

fields as well as the error estimates. In the error analysis, actual mismatch values from a fraction of withheld measurements are compared to the formal error estimate (Sect. 3.2.3). In the appendix, interested readers find a brief discussion of the viscosity parameter (Appendix A) and a sensitivity assessment with respect to changes in SMB, surface geoemetry and to the flux correction term (Appendix B).

#### 4.1.1   Ice flux

For Vestfonna ice cap, the ice-flux field is very instructive (Fig. 3). For many drainage basins, ice flux is small near the ice divide and gradually increases downglacier. The increase stems from ice accumulated along flow lines as well as from flow convergence towards the lateral margin. Often, ice flux is highest at or below the equilibrium line altitude. For Gimlebreen, Frazerbreen, Aldousbreen, Søre and Nordre Franklinbreane, ice flux remains elevated up to the marine ice fronts. For Gimlebreen, these high values are explained by an increasingly positive apparent flux divergence $\dot{a}$ towards the ice front (Fig. 2e).

Also for Aldousbreen, $\dot{a}$ stays positive near the glacier tongue. Unlike these examples, the apparent flux divergence turns negative long before the margin is reached for Nordre Franklinbreen and Frazerbreen. There, elevated ice-flux values are maintained by strong convergence. For Nordre Franklinbreen, the ice flux mainly follows the northern branch, whereas the 2015-2016 velocity information indicates that the southern branch is currently more active. The reason for this ice-flux deflection is that Franklinebreen was primarily drained through the northern branch in the 1990s (Schäfer et al., 2012) and that this route is still

dominantly imprinted in the surface geometry. For most of the ice cap, flux values decrease towards the margin and level out to zero in land-terminating areas.

For WSB and THPB, the ice flux is small all along the land-terminating margin and increases towards centrelines. For Austre Torellbreen, we find strong flux convergence along Bøygisen and Løveisen. Further downstream, ice-flux magnitudes remain constant as the apparent flux divergence is close to zero. Unlike this balanced situation, a large surface lowering signal on

Paierlbreen remains unexplained by the SMB, resulting in a positive apparent flux divergence over the entire catchment area. This imbalance is compensated by extensive downwasting implying a gradual flux increase up to the marine ice front. The





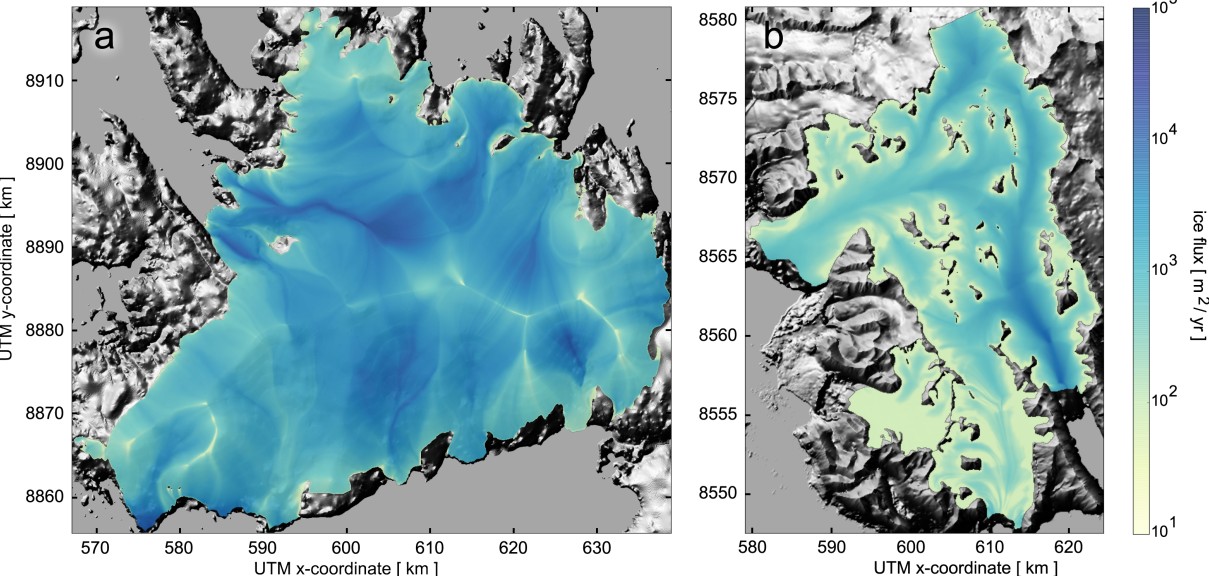

**Figure 3.** Ice-flux solution after cost optimisation for VIC (a), THPB and WSB (b). The ice flux gradually increases and converges into several distinct outlet glaciers. Flux values at marine ice fronts are part of the solution whereas a zero condition is prescribed along the land-terminating margin. Background: grey-scale hill-shaded topography based on a NPI 50m DEM.

imbalance itself might partially reflect the long-term geometric adjustment of Paierlbreen to the surge in 1993-1999. Yet, we cannot exclude that the SMB model underestimates the magnitude of surface melting or that a bias is introduced by the DEM differencing (Sect. 2.6). In any case, the Paierlbreen setup is challenging because there is almost no sink in the apparent flux divergence implying that ice mass is largely lost via the marine ice front. The tongue of Hansbreen shows less elevated flux

values instead which are explained by the small source area ($\dot{a} > 0$ in Fig. 2f). The situation is even more extreme on WSB, where such source areas are mostly limited to two small glacier branches joining the main branch from the north. Though they provide a certain in-flux, values along the main branch remain close to zero. Under the imposed SMB and elevation-change fields, no important ice-dynamic balancing is needed. Accepting a certain uncertainty in the imposed fields, we expect that WSB is stagnant at lower elevations because of the general glacier retreat on Svalbard.

## 4.1.2   Ice thickness and bedrock elevation

The first-step thickness map (Fig. 4) depends on surface slopes, thickness measurements and the ice-flux solution. The latter reflects both climatological and geometric information. For VIC, we find a mean thickness of 226m (Table A1). This value is significantly higher than the previously reported 185m, which was inferred from a direct kriging interpolation of the observations (Pettersson et al., 2011). One reasons for differences is that our reconstruction produces thicker ice along outlet

glaciers troughs. Such deep and often over-deepened channels (Frazerbreen and Franklinbreen in Fig. 5) are explained by con-



**Figure 4.** Ice thickness map for VIC (a,c) and THPB/WSB (b,d) as suggested by the first-step reconstruction approach. Thickness values for marine ice fronts are non-zero and a natural outcome of the underlying mass budget calculation. For VIC, thickness measurements (coloured dots) were collected with airborne radio-echo sounding instruments (Dowdeswell et al., 1986) as well as with ground-based pulsed radar systems (Pettersson et al., 2011; Navarro et al., 2014). For THPB/WSB, measurements were collected during several GPR campaigns between 2004-2012 (Navarro et al., 2014). The upper (a,b) and lower (c, d) panels show the respective thickness fields when all or only 1% of all thickness measurements were used in the first-step reconstruction, respectively. Background: grey-scale hill-shaded topography based on a NPI 50m DEM.



vergent ice flow draining large zones of the ice-cap accumulation area (Dowdeswell and Collin, 1990). For Braggebreen and Gimlebreen, the reconstruction suggests deep troughs which arise from a very positive, apparent flux divergence. The troughs are absent in the kriging interpolation as no observations were collected in this region. Another reason for differences is that kriging is expected to underestimate the ice thickness along the land-terminating margins away from observations because of

ice-free conditions outside the domain. For our approach, margin thicknesses are affected by physical quantities such as ice flux and surface geometry. An illustrative example for this effect is the dome-like surface topography of Forsiusbreen in the southwest of VIC (Fig. 2a). This glacier is almost deconnected from the main ice cap and the closest thickness measurements were taken more than 10km away. As a consequence, Pettersson et al. (2011) generate limited thickness values from kriging. In our reconstruction, a small ice dome is predicted (Fig. 4a) that is even grounded slightly below sea level in its central areas

(Fig. 5a). In addition, the thickest ice is no longer suspected beneath the main crest but our reconstruction suggests a maximum east of Frazerbreen, where values locally exceed 450m. In general, the first-step thickness map suggests that more than 12% of the ice-covered area is grounded below sea level. Previously, it was thought that only a 5%-area fraction lay below sea-level, due to limited measurements from the outer part of the ice cap. In terms of total ice volume, the first-step thickness map yields $534.7\,\mathrm{km}^3$ as compared to the $442\,\mathrm{km}^3$ from kriging (Pettersson et al., 2011).

For the THPB and WSB systems in southern Spitsbergen (Fig. 4b), an abundant observational record was available. Therefore we expect that relative differences between thickness maps from a direct interpolation and the first-step reconstruction should be small. From a direct kriging interpolation by Navarro et al. (2014), the mean thickness estimate for the THPB system is 184 m as compared to 182 m, here (Table A1). For the land-terminating WSB, mean thicknesses of 119 and 109 m are found, respectively. Relative differences in these values are small with 1.1% for THPB and 8.4% for WSB. The slightly updated vol-

ume estimates are then $55.5\,\mathrm{km}^3$ and $2.9\,\mathrm{km}^3$, respectively. Despite the similarity in these values, we see several systematic differences in the thickness maps from these two approaches. First, the kriging map shows that the measurements were interpolated ignoring the presence of some ice-free nunataks (for example above the confluence of Bøygisen and Løveisen in Fig. 4 in Navarro et al., 2014). Similarly, ice thickness does not tend to zero along some land-terminating margins. These positive biases are compensated in other areas, where thickness measurements are not reproduced after kriging. A clear difference is

seen along Vrangpeisbreen (Fig. 2b). In its upper reaches, the direct interpolation shows values below 100m (Fig. 4 in Navarro et al., 2014), whereas the thickness measurements along the centreline readily exceed 200m (Fig. 4b). These measurements are by construction reproduced here. Turning to the basal topography, we find elongate troughs reaching far upglacier from the marine terminus (Fig. 5b). The bedrock elevation is below zero over 14% of the entire THPB area. For Hansbreen, the bed remains below sea-level almost up to Kvitungisen (Fig. 2b). Even up at Nornebreen, a main tributary of Paierlbreen, the ice is

grounded well below sea-level.

    We re-computed the VIC, THPB and WSB thickness fields relying on a random 1%-sample of all thickness measurements (Fig. 4c,d). The idea is to assess the consequences of a lack of in situ measurements. For many glaciers around the globe, only few or often even no measurements are available. We therefore briefly discuss consequences from a lack of measurement constraints. For VIC, we find somewhat reduced values for the mean ice thickness of 216m and the total ice volume of $510.6\,\mathrm{km}^3$

(Table A1). Despite this reduction, general characteristics of the basal topography are already imprinted in the poorly informed



**Figure 5.** Bedrock topography associated to the thickness field in Fig. 4 for VIC (a,c) and THPB/WSB (b,d). Upper and lower panels reflect the respective amount of considered thickness measurements as in Fig. 4. Ice-free background: grey-scale hill-shaded topography based on a NPI 50m DEM. Ice-covered background: grey-scale hill-shaded bedrock topography.

reconstruction (Fig. 5c). Along the central survey track at the northern branch of Nordre Franklinbreen (Fig. 6a,b,c,d), the reconstruction provides an acceptable estimate of the withheld thickness measurement profiles. For THPB and WSB, the mean ice-thickness values are reduced to 129 and 108m from previously 182 and 109m, respectively. For THPB, the relative thickness reduction is substantial with 29%. As a consequence, this reconstruction only finds a 5% area fraction being grounded below sea level as compared to 14%, before. In many places, the sparsely informed reconstruction underestimates the depth



**Figure 6.** Ice thickness (a,c,e) as in Fig. 4 and bedrock topography (b,d,f) as in Fig. 5 for Nordre and Søre Franklinbreen on VIC. To facilitate visual comparability, the outline of the subdomain of Franklinbreen that is updated in the second-step reconstruction is highlighted (non-transparent lurid colours). First (a,b) and second rows (c,d) show the inferred geometries using 1% or all thickness measurements during the reconstruction respectively. The last row depicts the updated geometry after the second-step. Partially transparent areas in these maps (unsaturated colours) stem from the associated first-step reconstruction.

of elongate, narrow bed troughs (e.g. Nornebreen, Vrangpeisbreen). Hansbreen is an ideal test case to estimate how well the reconstruction performs without much thickness constraints because it is there that the GPR survey net is densest. Withholding 99% of the thickness measurements, the reconstruction suggest an elongate deep trough. Considering all measurements, the trough is actually somewhat deeper (especially close to the divide with Vrangpeisbreen) and ice-covered valley sides are steeper. Moreover, ice in tributary glaciers is found to be thicker. Despite the magnitude, the well-informed thickness reconstruction shows more spatial variability in central areas, which is not produced when almost no thickness measurements are

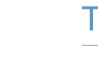
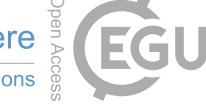

considered. This implies that small-scale features in the bedrock topography are not necessarily well imprinted in the glacier surface. Admittedly, a certain degree of details has been removed by a-priori smoothing of the surface topography.

### 4.1.3 Thickness error estimates

The following error analysis is two-fold: we first present and discuss error-estimate maps from the formal error propagation of
input uncertainties as described in Sect. 3.2.3. Secondly, we split the abundant thickness measurement record into two subsets. One subset is used in the reconstruction (Sect. 3.2.2), whereas the remainder is withheld for validation. The validation subset is used to infer actual mismatch values at the respective measurement location. Average values for the actual mismatch are then compared with the respective formal error estimates.

### 4.1.4 Estimates from error propagation

Relying on a formal error propagation (Sect. 3.2.3), it becomes possible to provide an error map (Fig. 7a,b). Using all thickness observations, the survey tracks are clearly discernible in all error maps. The 5m-constraint along these tracks (Lapazaran et al., 2016) is propagated along flux streamlines both upstream and downstream. Consequently, error estimates only gradually increase along flow, whereas more abrupt variations appear perpendicular to the inferred flux direction. Moreover, error estimates are highest in areas where ice flux is small as, for example near unconstrained divides and on a large portion of WSB. We
therefore suggest that future measurement campaigns should give priority to across-flow profiles as well as to stagnant areas. From gaps in the available cross profiles, we anticipate that a measurement spacing appropriate for thickness reconstructions should be in accordance with the target thickness map resolution. Once measurement gaps exceed the nominal resolution, error streaks start propagating through cross profiles. The by-far largest error estimates amongst all test geometries are found for the land-terminating WSB. These extreme values are caused by negligible ice flux over a major part of the domain (Sect. 4.1.1).
The error-estimate map also highlights that measurements should ideally be acquired on both sides of an ice divide. For Idunbreen (Fig. 2a), no measurements were obtained (Fig. 4a), which leads to elevated error estimates over most of this drainage basin (Fig. 7a). Thickness measurements collected just across the ice divide were not transmitted over the crest to the Idunbreen catchment area.

Considering only 1% of all thickness measurements, the error estimates become larger (Fig. 7c,d). In this case, the ice-cap
setup shows largest errors along the ice divide. The increase in the error estimates are unproportional with respect to the actual differences in the thickness values (Fig. 4a,c). Therefore, the fact that error estimates exceed the actual thickness value near the ice divide should not be overrated. For the valley glaciers THPB and WSB, maximum error estimates are found in areas with small ice flux. Errors are most prominent for WSB and Hansbreen, which both were found to exhibit small or even negligible ice flux. In addition, errors tend to be higher along central flow lines as a result of convergent ice flux. For sparse measurements
on ice caps and glaciers, we confirm that local error estimates readily reach 50% of the inferred thickness values.





**Figure 7.** Error-estimate map based on the error propagation presented in Sect. (3.2.3) for VIC (a,c) and THPB/WSB (b,d). Error estimates are equal to the 5m measurement uncertainty where observations were collected. Upper and lower panels reflect the amount of considered thickness measurements as in Fig. 4. Background: grey-scale hill-shaded topography based on a NPI 50m DEM.

### 4.1.5 Actual thickness mismatch

A pressing question is whether the magnitude of these error-estimate maps is reliable and falls into a realistic range. For this purpose, we withheld a random sample of all thickness measurements from the reconstruction and computed an absolute



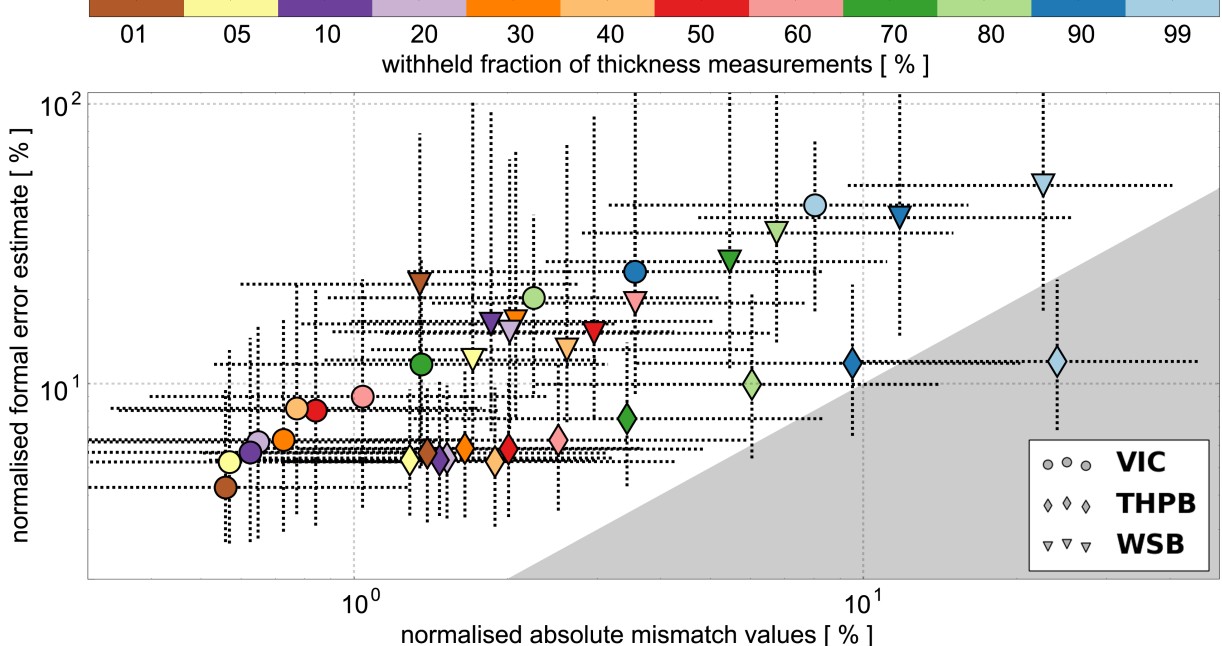

**Figure 8.** Normalised median values for the absolute thickness mismatches and the error estimates at measurement locations not included during the reconstruction. Medians are normalised to the average thickness value of the withheld measurements. Marker colours indicate the respective fraction of all measurements withheld from the reconstruction. Dashed crosses span the interquartile range of all mismatch values (horizontal) and all formal error estimates (vertical). For orientation, the grey background shading was added to highlight the identity line.

thickness mismatch for comparison. The sample size is defined as a fraction of all measurements and we investigated the range from 1 to 99%.

In a first attempt, we directly compared the formal error estimates to the in situ absolute mismatch values. Ideally, these two values would show a positive correlation. Yet, no clear dependence was discernible for none of the sample sizes. Both data distributions, for mismatch values and error estimates, are not normal and we therefore decided to quantify them in terms of medians and quartiles. These measures are more robust to outliers than mean values and standard deviations. Medians and quartiles are normalised to the average thickness of all withheld observations (Fig. 8). First and foremost, medians and quartiles suggest that error estimates tend to overestimate the absolute mismatch. For small fractions of withheld measurements, the overestimation is stronger. This bias does not surprise as formal error estimates cannot fall below a 5m limit Lapazaran et al. (2016), whereas high correlation between thicknesses at adjacent location results in very low mismatch values. If only 1% of the measurements is withheld, normalised medians of the absolute mismatch range from 0.5 to 1.4%, whereas equivalent values for the error estimates range between 4 and 22%. As more and more data is withheld, normalised median mismatches increase. For a withheld data fraction of 99%, we find values of 8% for VIC, 24% for THPB and 23% for WSB. These values could give a first indication of the maximum overall uncertainty associated with the presented thickness reconstruction for glaciers for

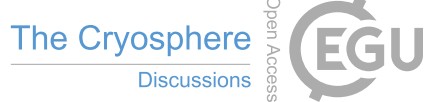



which no measurements are available. The normalised median error estimates are 43%, 11% and 51%, respectively. Again, WSB stands out here because formal error estimates diverge over a large portion of the glacier area where flux values are very small. From this comparison, we take that formal error estimates show a tendency to exceed mismatch values. This tendency suggests that error estimates, here, can be interpreted in terms of an upper constraint. Admittedly, magnitudes of these error

estimates depend on the choice of te input uncertainty, which will affect the interpretation. We strongly want to discourage that this error estimates are interpreted in terms of a standard deviation because of two reasons. First, mismatch values and error estimates are not correlated and, secondly, both distributions are not normal.

## 4.2 Second-step reconstruction

The second step of this reconstruction is optional and depends on the availability of velocity information. Knowing surface

velocities, the mass conservation equation can be solved directly for the unknown ice thickness (Morlighem et al., 2014). The second-step thickness field is anticipated to be an improvement. The reason is that the flow direction is no longer geometrically prescribed but follows the observed surface velocity field. In addition, the pattern of velocity magnitudes enters the reconstruction and modulates the thickness field accordingly. We decided to limit the thickness update to areas in which the observed velocity magnitude exceeds 100 m yr$^{-1}$. Below this threshold, the velocity information becomes increasingly

fragmented (Fig. 2g,h).

### 4.2.1 Ice thickness

On VIC, ice thickness is updated along 8 fast-flowing outlet glaciers (Figs. 6e and 9a,c). In these areas, the new thickness field can differ considerably from the first-step reconstruction, particularly in areas with sparse observational constraints as for Idunbreen and Rijpbreen. The reason is that velocity streamlines deviate from the slope-prescribed flux direction. Consequently,

the ice is distributed differently. For Idunbreen and Rijpbreen, deeper troughs are found somewhat away from the ice front which is explained by a convergent surface velocity field. Therefore, velocity measurements seem more valuable in areas where no thickness measurements are available. Along the southern branch of Nordre Franklinbreen, some measurements were collected, constraining the reconstruction. The updated thickness field is anyhow thicker downstream of the junction between the southern branch and the not-updated northern branch (Fig. 6e). The updated bedrock topography now shows a somewhat

deeper trough (Fig. 6f). The reason for this difference is that ice flow follows the 2014-2015 surface velocities, which favoured the southern branch, while the geometrically imposed first-step flux direction showed a preference for northward outflow (Fig. 3).

In Wedel Jarlsberg Land, thickness fields are updated for three fast-moving frontal areas of each glacier in the THPB complex. The wealth of thickness observations implies that the first- and second-step reconstructions are very similar (Fig. 9b).

This is certainly the case for the fast portions of Hansbreen and Paierlbreen. Differences become largest near the calving fronts because of the free boundary condition. For Hansbreen, the bed trough near the ice front becomes both deeper and wider whereas the updated Paierlbreen geometry only shows a wider trough. For Austre Torellbreen, differences are more apparent





**Figure 9.** Ice thickness (a,b) as in Fig. 4 and bedrock topography (c,d) as in Fig. 5 for VIC (a,c) and THPB (b,d). Partially transparent areas in these maps (unsaturated colours) stem from the first-step reconstruction, for which values are inferred from the apparent flux solution. Along the outlet glaciers (non-transparent lurid colours), the two fields were updated accounting for velocity observations in the mass conservation.

as only two along-flow measurement profiles constrain the thickness field at low elevations. Along the centreline of Austre Torellbreen, two overdeepened spots in the first-step reconstruction are flattened out in the updated basal topography (Fig. 9d).





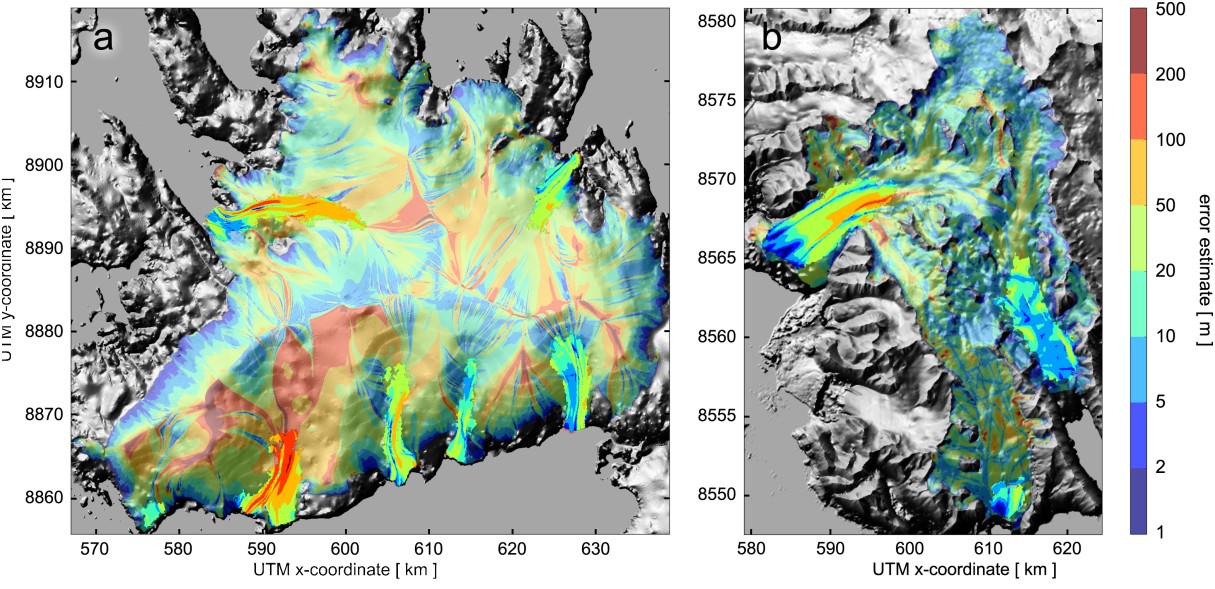

**Figure 10.** Error estimate map as in Fig. 7 for VIC (a) and THPB (b). Partially transparent areas in the thickness maps (unsaturated colours) stem from the first-step reconstruction, for which ice thicknesses are inferred from the apparent flux solution. Along the outlet glaciers (non-transparent lurid colours), error estimates are updated relying on velocity observations (Fig. 2c, d).

### 4.2.2 Error estimates

The updated error-estimate fields are informed by first-step values at lateral boundaries except for marine ice fronts (Fig. 10). Owing to this inheritance, a repetition of the above error analysis (Sect. 4.1.3) seems redundant . In the subdomains, error estimates are simply propagated along velocity streamlines and further modified. For VIC, magnitudes of the updated error estimates tend to increase as compared to the first-step values. This might imply that the input velocity uncertainty of $20\,\mathrm{m\,yr^{-1}}$ was chosen relatively high. Yet, this tendency is not confirmed for the THPB complex. Along the survey tracks, error estimates are identical as we again prescribe the 5m value associated to the thickness measurements (Lapazaran et al., 2016).

## 5 General Discussion

In this section, we discuss the central assumptions and caveats of the presented reconstruction approach. For the first step, sliding is inherently neglected, assuming that ice motion is an exclusive result from internal deformation. In areas without thickness and velocity information, this assumption is likely the dominant source of uncertainties and might even bias the results towards high thickness values. Other reconstruction approaches use an empirical scaling relation (e.g. Farinotti et al., 2009b) or incorporate a transiently resolved relation for basal water availability (van Pelt et al., 2013). In either case, formulations are basic because of our limited knowledge of basal conditions. Although these approaches are valuable attempts to address the



issue of unknown basal conditions, it remains questionable whether uncertainties in the reconstructed thickness field are in fact reduced. Here, we instead address basal sliding by relying on direct measurements of the surface velocities but limited to sub-domains where magnitudes exceed $100\,\mathrm{m\,yr^{-1}}$. These measurements comprise motion arising from both internal deformation and basal sliding.

5 Another caveat in the first-step reconstruction is the assumption that ice flux follows the steepest surface slope (Sect. 3.2). Although this assumption might be appropriate in slow-moving areas, the actual velocity vector can point into a different direction. The situation becomes even more complex for surging glaciers, for which the surface topography is significantly modified during these short-term events. An examples is Franklinbreen on VIC. Here, the geometrically imposed flux direction prefers the northern outlet branch (Fig. 3). Although this is consistent with velocities in the early 1990s, the 2015 state shows

10 that more ice is currently exported via the southern branch. Therefore, the surface topography is not necessarily the best indicator for the flow direction. In the second step, we were able to update the thickness field in consistency with the recent velocity fields. Yet even for the second-step reconstruction, it is not evident how to account for important, non-regular dynamic changes, such as surging, as for instance on Franklinbreen and Paierlbreen (Błaszczyk et al., 2009).

From the perspective of mass conservation, uncertainties from temporal inconsistencies could be reduced by contemporane-

15 ous input fields. If some input fields are representative for very different time periods than others, a not-well assessable bias is introduce in th reconstruction. Despite time consistency, input fields should also be averaged over a certain period. The reason is that input variables can show strong seasonal and inter-annual variability, which alter the inferred thickness fields. This effect was also seen for Franklinbreen for which ice preferred different export paths in the two reconstruction steps. There-fore, some inherent response time of glacial systems has to be accounted for in the averaging. For the test cases described in

20 this manuscript, time consistency could not be prioritised because of limited data availability. Resultant time inconsistencies therefore add to the uncertainties associated with the presented thickness fields.

Concerning the sensitivity of the thickness map of VIC to changes in the input fields (Appendix B), we find that integrated values as mean ice thickness and ice volumes are rather insensitive. On VIC, relative differences in our analysis remain below 3% (Table A1). Differences in these integrated values reduce as more and more thickness measurements are available. Lo-

25 cally, differences can however become large and the explanation is not always evident. Without thickness measurements for correction, we found that an offset in the specific SMB directly translates into a thickness bias. Concerning the flux correction, we confirm that it is most influential for stagnating areas. Ice flux values on Werenskioldbreen are very small and the relative volume difference when applying the flux correction reaches 8%. Yet the effect reduces both with the availability of thickness measurements and with an increasing mass overturning as prescribed by the apparent flux divergence. For VIC, ice-volume

30 or mean-thickness estimates change by less than 0.3% after applying the flux correction. For THBP, the relative difference is slightly larger at 2.4%. For these cases, these relative differences are comarably small as compared to the sensitivity to mea-surement availability. Using either all or only 1% of all available measuremetns in the reconstruction results in larger relative changes in the mean ice thickness by 5% for VIC and 29% for THPB.

For the second-step in this reconstruction, we found that it is impractical to use the velocity observations over entire drainage

35 basins. The primary reason is that the reconstruction approach is very sensitive to inconsistencies in areas where velocity



magnitudes are small. Therefore, we decided to spatially limit the reconstruction using a $100\,\mathrm{m\,yr^{-1}}$ velocity threshold. This choice ensured that fragmented areas in the velocity field were avoided and that relative input uncertainties remain below 20%. Another reason was that the ice thickness reconstruction can only perform well where ice flow is linked to a designated source area. Sources can either be an area of dominant surface accumulation or an important upstream inflow boundary. When

applied to entire drainage basins on VIC, some isolated flow systems (e.g. the northern branch of Nordre Franklinbreen) were not linked to any source area. As a consequence, the velocity informed thickness reconstruction produced almost no ice cover in these areas. We are convinced that the second-step reconstruction could be extended to larger sub-domains. We think that such an extension is possible by lowering the velocity threshold while assuring that most velocity streamlines connect to an upstream inflow boundary. Yet, the actual choices involved are not evident and need further investigation which exceeds the

scope of this study.

## 6 Conclusions

We have presented a two-step, mass-conserving reconstruction approach to infer ice thickness maps with prior knowledge on source and sink terms in the mass budget. The first step is intended for glaciers for which input information is limited. Requirements are comparable to other reconstruction approaches that have been applied successfully to glaciers world-wide

(Huss and Farinotti, 2012). In fast-flowing areas, available velocity information is used, in a second step, to improve the thickness field along outlet glacier sub-domains. In both steps, available thickness measurements are readily assimilated to constrain the reconstruction. The approach is tested on different glacier geometries on Svalbard where an abundant thickness record was available. On these test geometries, we show that the approach performs well for entire ice caps as well as for marine- and land-terminating glaciers.

For the land-terminating Werenskioldbreen in southern Spitsbergen, measurement tracks are densely spaced. Therefore, average thickness values from a direct interpolation (Navarro et al., 2014) and from the first-step reconstruction are very similar at 119 and 109m, respectively. Dense measurement tracks were also acquired on the adjacent Austre Torellbreen, Hansbreen and Paierlbreen complex for which the average thickness value of 186m deviates by mere 2% from the direct interpolation. However, reconstructed thickness values along the land terminated margin are somewhat smaller than from

the direct interpolation whereas ice tends to be thicker along central flow lines and away from constraining observations. For the three marine-terminating glaciers, the mean ice-front thickness is just above 100m, which is loosely confirmed by the rough archipelago-wide estimate of 100m which was necessary as no measurements or reconstructions were available (Błaszczyk et al., 2009). The thickness field of an ice cap is considered a more challenging task for reconstruction because of an increasingly flat topography near the ice divide. For Vestfonna ice cap, we find a mean first-step thickness value of 226m,

about 22% larger than the previously reported glacier-mean of 186m (Pettersson et al., 2011). In addition, the fraction of ice grounded below sea-level needs substantial upward correction, from 5% to 14%. Consequently, ice loss under future climatic warming will be intensified by iceberg calving to the surrounding oceans over a much larger area. In the second step of our approach, the ice thickness field was updated using ice velocity measurements over prominent marine-terminating glaciers.





The resultant thickness field is more consistent with the actual ice-flow pattern and therefore considered an improvement. In areas without thickness measurements, the second step reconstruction can produce thicker ice in confluence areas.

The reconstructed thickness field is provided together with an error-estimate map, which stems from a formal propagation of input uncertainties through the underlying equations. For the first-step reconstruction, the magnitude of these error estimates was validated against mismatch values computed from withheld measurements. We find that formal error estimates tend to overestimate mismatch values. Analysing their distribution, error estimates can here be considered upper and lower constraints of inferred thickness values. The error-estimate maps highlight that survey tracks should preferentially be planned in across-flow direction. The error map is also valuable as it points out the regions that were thickness values are least well constrained and should therefore be target areas when planning future surveys. Generally error estimates diverge over ice divides and in stagnant areas. The fact that we withheld fractions of all thickness measurements could further be exploited in terms of a first overall uncertainty estimate for glacier for which not many or even no thickness observations are available. From the analysis of resultant mismatch values, we expect a glacier-wide median uncertainty in the reconstructed thickness field of about 25%, normalised to the mean glacier thickness.



## Appendix A: Viscosity parameter

To translate the ice-flux solution into an ice-thickness field, the ice-viscosity parameter $B$ has to be defined (Fig. A1). Parameter values are inferred at locations where thickness measurements are available via Eq. (5). The resultant point information is then interpolated over the entire glacier domain (Sect. 3.2.2). For VIC, we find values covering a spectrum from 0.02 to 0.54

5  MPa yr$^{1/3}$, which corresponds to a rate-factor range from $1.97 \cdot 10^{-25}$ to $1.98 \cdot 10^{-21}$ Pa$^{-3}$s$^{-1}$. For ice temperatures between -20 and 0°C, we would expect rate-factor values between $1.0 \cdot 10^{-25}$ and $2.4 \cdot 10^{-24}$ Pa$^{-3}$s$^{-1}$ (e.g. p. 75, in Cuffey and Paterson, 2010). The inferred values for VIC clearly exceed this meaningful range and should therefore not be interpreted in terms of a material property. The ice viscosity is a tuning factor, which compensates for any deficiencies or inconsistencies in the reconstruction. The parameter is also affected by the flux correction in stagnant areas. The highest viscosities are inferred

10  in areas next to land-terminating boundaries. These areas are also characterised by small flux values. As observations show some non-negligible thickness values there, $B$ has to be high. The lowest values are seen in the northern part of the ice cap and along the lower trunk of Aldousbreen. For this glacier, one might interpret these low values in terms of sliding. However, for other outlet glaciers, the viscosity parameter is not necessarily decreased as compared with the surrounding area. This inconsistency also suggests that a physical interpretation of the viscosity parameter is delicate. For the THPB and WSB area,

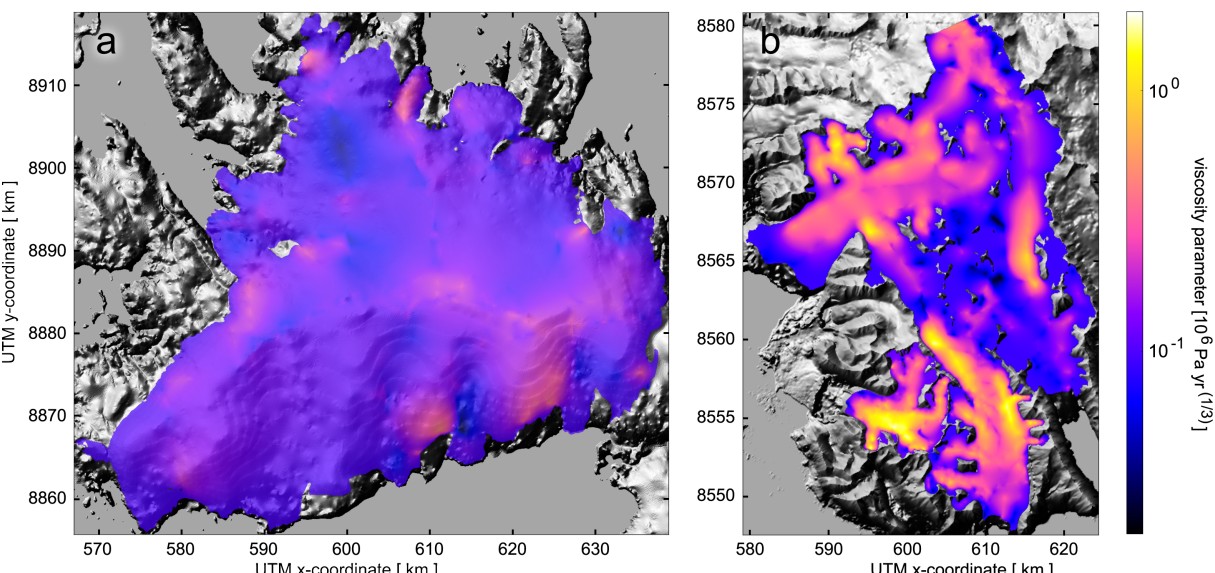

**Figure A1.** The ice-viscosity parameter $B$ for VIC (a) and THPB/WSB (b) is inferred during the first-step of the reconstruction as explained in Sect. 3.2.2. This parameter is used to match observed and reconstructed thickness values. An interpretation in terms of material property is delicate because the parameter compensates for input uncertainties and inconsistencies as well as for assumptions in the first-step reconstruction. Background: grey-scale hill-shaded topography based on a NPI 50m DEM.

15  the $B$-field also shows strong variations (Fig. A1). Values cover a range from 0.03 to 1.95 MPa yr$^{1/3}$, corresponding to a rate





factor range between $4.26 \cdot 10^{-27}$ and $2.61 \cdot 10^{-21}$ Pa$^{-3}$s$^{-1}$. The inferred range is even larger than for VIC and exceeds the physical range. Yet for these glaciers, a pattern might be discernible. High viscosities are often concentrated along central glacier flowlines. One explanation could be that the flux solution shows a low bias along these trunks as a result of systematic inconsistencies between the input SMB and the surface elevation changes. Such a systematic effect would naturally cumulate

as ice flow converges towards centrelines. This explanation is certainly supported on Hansbreen and Werenskioldbreen. There, the ice-flux solution shows comparatively small magnitudes, which likely explains unproportionly elevated viscosity values. Lowest viscosity values are concentrated along the ridges and in the flat area between the nunataks separating Paierlbreen and Austre Torellbreen.

In summary, the interpretation of this viscosity field $B$ in terms of ice dynamics is rather limited because values exceed the

physical range. The field should rather be seen as a multiplier for tuning purposes as it can compensate for uncertainties in and inconsistencies between input fields as well as for assumptions within this first-step reconstruction. $B$ is presented here to visualise that a single viscosity parameter might not be sufficient to capture all spatial variations in the thickness field. Initially, a best-fit single viscosity value over entire drainage basins was used, but the thickness pattern could not be explained by variations in ice flux and surface slopes alone (Eq. 5). A single viscosity parameter resulted in underestimated thicknesses

for the thick parts of the glacier and overestimated values for shallower parts (not shown). Other comparable state-of-the-art approaches often use a constant value for entire glacier basins (Farinotti et al., 2009b; Huss and Farinotti, 2012; van Pelt et al., 2013).





## Appendix B: Sensitivity analysis

### B1    Surface mass balance

Here, the sensitivity of the first-step reconstruction to the SMB input is briefly discussed for VIC (Fig. 9). For this purpose, we exchange the 1975-2015 MAR-SMB with the 2003-2013 WRF-SMB (Sect. 2.5). A fundamental discrepancy between the

simulated time periods becomes apparent when integrating the SMB fields over the ice cap. We obtain mean SMB values of -0.08 for MAR and -0.3 m i.e. $\mathrm{yr}^{-1}$ per unit area for WRF. For the WRF-SMB, more ice is removed at low elevations consistent with the warmer climatic conditions of the more recent period. When using all thickness measurements, the new thickness field (Fig. A1a) is very similar (Fig. 4a) showing a slightly reduced mean value of 222m (Table A1). Consequently, the new volume estimate is also reduced to $525.2\,\mathrm{km}^3$ (about 2%). Reduced thickness values are visible near the ice fronts

of Gimlebreen, Idunbreen and Bodleybreen. Due to a lack of observations in these regions, the reconstruction is not well constrained and as the WRF-SMB removes more ice, glacier thickness estimates become smaller. This reduction is important as the ice cliff height determines the unknown ice discharge. The frontal reduction is less clear for the land-terminating margin because steeper surface slopes limit the ablation-zone extent. The reduction becomes even more evident when only 1% of the thickness measurements is used (Fig. A1b). Thickness values near the ice divide are however not necessarily smaller. On

average, the ice volume estimate is reduced to $497.9\,\mathrm{km}^3$ and a mean thickness value of 210m is found (-3%).

In general, the reconstruction is capable of compensating poorly constrained SMB data where the thickness record has high spatial coverage. For glaciers where no information is available neither on ice thickness nor on surface mass balance or elevation changes, the reconstruction is largely unconstrained. Without thickness information, the error estimates can be reduced by investing in consistent, contemporaneous and well-informed fields for SMB and surface elevation changes.

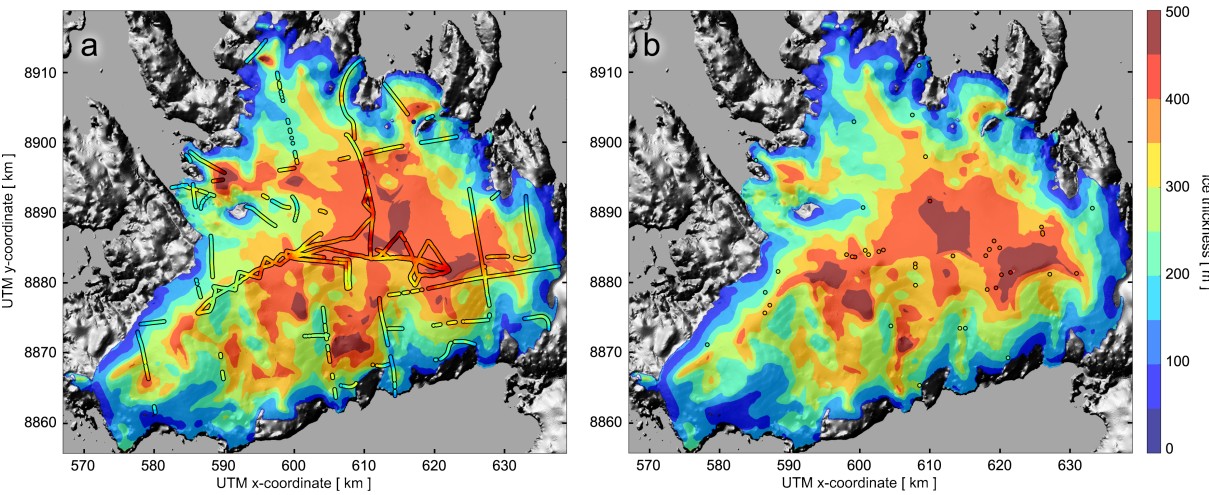

**Figure A2.** Ice thickness map for VIC as in Fig. 4 based on the 2003-2013 WRF-SMB field using all (a) or only 1% (b) of the thickness measurements.





## B2  Surface topography

The sensitivity of the first-step thickness field to the DEM choice is somewhat smaller for VIC. The exchange of the 2010 DEM (Sect. 2.3) with the NPI 1990 DEM results in mean thickness and ice-volume reduction of less than 3% (Table A1). The difference reduces below 1% if all thickness measurements are used during the reconstruction. Moreover, the reduction in maximum thickness values is comparably larger with more than 10%. This latter reduction is symptomatic for an overall less variable thickness field because the NPI DEM was computed from contour lines and is therefore rather smooth. In the ice thickness map (Fig. A3), there are many small changes in the pattern. One more prominent difference is that the lower trunk of Franklinbreen becomes more elongate and deep. Pattern differences are again more expressed in the case that less thickness observations were used. Locally relative thickness differences can become very high. Therefore, the DEM choice is certainly important for inferring local thickness values. Yet relative differences in the total ice-volume estimates are small as compared to expected mismatch values of more than 25%, if no observations were available.

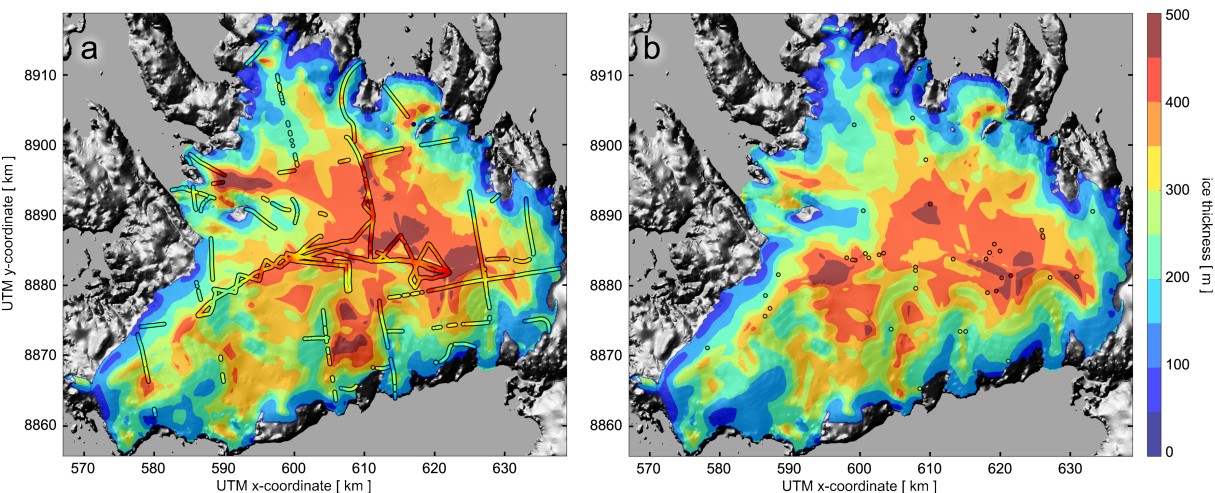

**Figure A3.** Ice thickness map for VIC as in Fig. 4. Here, the reconstruction is conducted with the NPI 50m DEM as surface topography using either all (a) or only 1% (b) of the thickness measurements.

## B3  Negative ice flux

The flux correction applied during the first-step reconstruction (Sect. 3.2.2) could be considered an important bias. Note however that, the correction is not added to the flux solution itself (Fig. 3) and that it does not enter the error calculations (Sect. 3.2.3). The correction is only applied when inferring ice thickness values for the purpose of avoiding zero transitions in areas where flux values turn negative. In this way, it only affect the ice thickness and the viscosity parameter. For VIC, negative flux values occur on a 0.5% area fraction. For THPB and WSB, this values is more elevated with 1.5 and 4.4%, respectively. In these areas, the flux solution and the geometrically imposed flux direction cannot be reconciled. The negative values prevail



**Table A1.** Reconstruction sensitivity as quantified by the mean and maximum ice thickness, the ice volume and the area fraction grounded below sea-level. The ‡-symbol separates values stemming from a reconstruction using either all or only a 1% fraction of the available thickness measurements.

| setting | glacier geometry abbr. | mean thickness thickness [ m ] | | | maximum thickness thickness [ m ] | | | ice volume [ km³] | | | area fraction below sea-level [ % ] | | |
|---|---|---|---|---|---|---|---|---|---|---|---|---|---|
| reference | VIC | 226.0 | ‡ | 215.8 | 452.6 | ‡ | 461.3 | 534.7 | ‡ | 510.6 | 12.4 | ‡ | 10.0 |
| | THPB | 182.0 | ‡ | 129.2 | 642.8 | ‡ | 564.9 | 55.5 | ‡ | 39.4 | 14.4 | ‡ | 4.77 |
| | WSB | 109.2 | ‡ | 108.0 | 378.3 | ‡ | 301.9 | 2.92 | ‡ | 2.89 | 0.27 | ‡ | 0.56 |
| WRF-SMB | VIC | 222.0 | ‡ | 210.4 | 467.2 | ‡ | 454.4 | 525.2 | ‡ | 497.9 | 10.7 | ‡ | 7.65 |
| NPI 50m DEM | VIC | 224.0 | ‡ | 211.1 | 425.2 | ‡ | 413.7 | 530.1 | ‡ | 499.4 | 12.2 | ‡ | 8.95 |
| no flux correction | VIC | 226.1 | ‡ | 215.2 | 490.6 | ‡ | 582.0 | 534.9 | ‡ | 509.1 | 12.5 | ‡ | 9.41 |
| | THPB | 180.8 | ‡ | 126.2 | 664.0 | ‡ | 598.7 | 55.1 | ‡ | 38.5 | 14.6 | ‡ | 6.38 |
| | WSB | 109.7 | ‡ | 116.5 | 408.3 | ‡ | 385.8 | 2.94 | ‡ | 3.12 | 1.45 | ‡ | 1.07 |

despite the penalty in the cost function during the optimisation (Sect. 3.2.1). An increase of the respective multiplier in the cost function resulted only in a limited improvement on WSB and came at the expense of a more variable flux field on all geometries. Therefore we rather decided to introduce a correction term that guarantees positive flux values in the SIA equation (Eq. 5). The correction is primarily required for WSB for which magnitudes of the flux solution are very small. Anyhow, we

applied it to all geometries to keep uniformity in the approach.

    Here, we want to present the thickness solution in the case that no flux correction is applied (Fig. A4). For VIC and THPB, differences in the thickness maps are spatially very confined and thus difficult to discern. For VIC, streak features with small thickness values appear for instance on Braggebreen (in the southwest). A similar feature is seen on Hansbreen just north of the confluence with Staszelbreen. More prominent are the effects on WSB. There, a noise pattern of near-zero values appears

on the thickness field of the main trunk where flux values are small (Fig. 3). The bogus noise pattern is not acceptable as we expect that the thickness field shows more gradual changes. For VIC, changes in mean ice thickness and ice volume remain below 1% (Table A1). This also holds for all test geometries when all available thickness measurements were used during the first-step reconstruction. Using only 1% of the thickness measurements, relative differences increase to 2.5% on THPB and 7.9% on WSB. For THPB, a reduction of ice volume is found without flux correction while thicker ice is predicted for

WSB. A welcome side-effect of the flux correction is a general decrease in the maximum thickness values which also appear in stagnant areas. In summary, the effect of the flux correction can lead to a considerable difference in ice volume in the case that no thickness measurements are available and that small flux values prevail over a large area. Yet, the correction results not necessarily in an increase of ice volume because of possible compensating changes in the ice-viscosity parameter. In addition, the effect of this correction is expected to be large for stagnating glaciers whereas for dynamically active glaciers, consequences



**Figure A4.** Ice thickness for VIC (a,c) and THPB/WSB (b,d) as in Fig. 4. Here, the first-step flux solution is not corrected to avoid negative flux values in the SIA-equation used to infer the ice thickness (Sect. 3.2.2). For WSB, you see that many patches appear with very small thickness values. These bogus variations are a consequence of zero transitions in the flux field. For VIC and THPB such bogus variations are limited to some few small areas.

will be negligible. The ice-flux field gives an indication on if consequences are expected to be large and where they will be most expressed. In any case, the error-estimate map will highlight areas in which this correction is important. For the main trunk of WSB, error estimates exceed by far the inferred thickness values (Fig. 7b).





*Author contributions.* J.J.F. designed and implemented the reconstruction approach, applied it to the test cases and elaborated the details of the error estimation. The research aims and setup was developed in regular discussion with F.G.-C., T.S., B.S. and M.B. J.J.F. led the writing of the manuscript, in which he received support from all authors. F.G.-C. developed and provided the initial version of the optimisation routines. Input fields for the reconstruction are Sentinel-1 surface velocities from T.S., ice thickness measurements from T.J.B., J.A.D., R.P., F.N. and M.G., DEMs from C.N. and B.S., surface elevation changes from C.N. and G.M., and surface mass balance fields from X.F.,C.L. and K.A.

*Competing interests.* The authors declare that they have no conflict of interest.

*Acknowledgements.* This study received primary funding from the German Research Foundation (DFG) within the Svalbard - iFLOWbed project, grant number FU1032/1-1. Results presented in this publication are based on numerical simulations conducted at the high performance computing centre of the "Regionales Rechenzentrum Erlangen" (RRZE) of the University of Erlangen-Nuremberg. The reconstruction approach also benefits from co-development work of the Elmer/Ice team at the CSC-IT Center for Science Ltd (Finland). The velocity analysis on Svalbard was funded by DFG within the priority programme 1158 Antarctic Research under contract number BR2105/9-1 and received financial support from the Helmholtz Association of the German Research Centres (HGF) Alliance on Remote Sensing and Earth System Dynamics.Thickness data collection in Wedel Jarlsberg Land was funded by the Spanish R&D projects C11093001 and C150954001, the NCBiR/PolarCLIMATE-2009/2-2/2010 from the Polish National Centre for R&D, by the IPY/269/2006 from the Polish Ministry of Science and Higher Education, by Polish-Norwegian funding through the AWAKE (PNRF-22-AI-1/07) project, by the EU FP7 ice2sea programme (grant number 226375) and by funds of the Leading National Research Centre (KNOW) received by the Centre for Polar Studies of the University of Silesia, Poland. The DEM generation in Wedel Jarlsberg Land received financial support from the European Union/ERC (grant 320816) and from ESA (project Glaciers CCI, 4000109873/14/I-NB). TanDEM-X data was provided under AO XTIGLAC6770. The WRF-SMB field was produced within the PERMANOR project funded by the Norwegian Research Council (255331).



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
