# Peer review of "Application of a two-step approach for mapping ice thickness to various glacier types on Svalbard"

_The Cryosphere, 2017_

## Short Comment (SC1) · 24 Mar 2017

The authors spend some effort on providing formal estimates on the overall accuracy of the method, which is greatly appreciated. I suggest that they also look into the detailed comparison of various radar processing schemes provided by Moran et al. (2000). Those authors compare raw 1D vs. 2D vs 3D array radar data processing and come up with estimates of ice thickness errors which are larger than those often used in other studies. Lapazaran, as the most recent and comprehensive study on this issue, cite it, too, but the original paper might be worth looking into, as your study also includes very mountainous valleys.

Delineation of a complexly dipping temperate glacier bed using short-pulse radar arrays

[Figure]

Moran, M. L.; Greenfield, R. J.; Arcone, S. A.; Delaney, A. J. Journal of Glaciology, Volume 46, Number 153, March 2000, pp. 274-286(13)

http://www.ingentaconnect.com/content/igsoc/jog/2000/00000046/00000153/art00012

---

## Referee Comment (RC1) · D. Brinkerhoff (Referee) · 28 Mar 2017

**Summary**

In this study, Fürst and others present an updated method for solving the problem of inferring ice thickness from sparse observations coupled with surface data. They then apply this method to three glacial systems on the Svalbard Archipelago. The validity of the resulting estimations of ice thickness are established with a detailed error analysis.

The paper is clearly relevant and within the appropriate scope for the journal. Contemporary interest in methods for inferring ice thickness are of great interest to the

community, as evinced by numerous recent publications on the subject, including a comprehensive intercomparison (Farinotti et al., 2016). This paper's contribution to the field stems primarily from its presentation of a way to circumvent some of the arduous data requirements required by the method on which it is based (Morlighem et al., 2010). I am concerned that this paper inherits some of the potential shortcomings from that work, namely an error analysis which is not, in my view, completely justified, as well as a misunderstanding of the definition of error for PDE-constrained optimization. Nonetheless, the manuscript does a commendable job with respect to discussing its own limitations and in firmly placing the issue in the context of error analysis. I am not sure that the paper will provoke a sea-change in thinking about Svalbard's glaciological processes, but I imagine that the results will be useful for modellers and others needing ice thickness estimates.

Stylistically, I think that the paper could benefit from significant distillation. The elimination of superfluous words, sentences, and perhaps even sections would help the reader to focus on essential points. As it stands, the manuscript feels like a methods paper mixed up with a case study. A stronger partitioning between these two parts would help. The paper also contains a fair amount of questionable English. I have tried to make corrections where I can, but a more detailed reading by the authors themselves is in order.

Ultimately, under the assumption that the authors can address the criticisms that I have presented below, I would encourage resubmission.

[Figure]

**1 Major Points**

**1.1 On the use of 'apparent flux divergence'**

The phrase 'apparent mass balance' is well ensconced in the literature at this point, and the reasoning for its use is fairly clear: when $\partial_t h$ is used as an observation, it acts identically to $\dot{b}$, which is a source term. Combining them leads to a simplified equation involving the flux divergence and this unified source term. The term 'apparent flux divergence' makes no sense at all. In fact, a more correct statement would be to just call the apparent mass balance the flux divergence (rather than apparent flux divergence), because that is what the equal sign implies. However, this would be confusing and tautological to say that the flux divergence equals the flux divergence. It is equally confusing, but perhaps less correct to say that the apparent flux divergence equals the flux divergence. If it's not clear already, I suggest using the term 'apparent mass balance' instead.

**1.2 General characteristics section, and the extensive use of proper nouns**

This paper walks a difficult line between a methods paper and a case study. I don't have a problem with that, as it's generally useful to see methods applied to real cases to evaluate their worth. However, I think that the organization of this paper is such that it can be quite confusing. I would suggest reorganizing the paper such that the methods are completely stated (including the theory behind the error analysis; more on that in a minute), then switch gears and begin discussing the nuances of Svalbard's geometry and data availability. Thus I would not have to remember what and where Austre Torrellbreen is after reading many pages about hyperbolic PDEs.

Additionally, all figures need to be labelled with salient features discussed in the text. The reader should not have to cross-reference Figure 2, while at the same time reading

the paper text and analyzing the figure content. Additionally, some basic annotations describing some particular key points referenced in the text and restated in the figure captions would be very helpful.

**1.3 Estimates of input observation error**

Input data error estimates need to be stated more clearly and with more complete justification. For example, the authors use a mean value of 5 m for estimated thickness error for all experiments, based on a (convincing) paper in which GPR was applied to a relatively thin glacial system, with lower wavelength antennae. It is not tenable to assume the same error estimate for airborne radar measurement from the 1980s. Also, no real effort is made to justify the 0.2 m yr$^{-1}$ estimate in apparent surface mass balance.

For all of these error sources, the distinction between simple observational error and the error associated with the variables used in your equations (which are time-averaged) needs to be discussed. See the methods section of Brinkerhoff et al. (2016) for a discussion of what I mean by this; in short, Eq. 1 assumes that all of the inputs are temporally consistent, but in reality they are not, and this induces an additional source of uncertainty.

**1.4 Flow directions**

I strongly suggest reading Kamb and Echelmeyer (1986), when approaching flow direction computations. They provide a strong theoretical basis for how to approach ice routing using only surface elevation observations in order to approximately recover higher-order directions. With the availability of these theoretical results, which have been used successfully many times throughout the balance flux literature, I find the *ad hoc* method of smoothing by applying a contouring algorithm then using natural neighbors interpolation to degrade the surface elevation accuracy to be troubling. Barring the success of the smoothing proposed by Kamb and Echelmeyer (1986) at eliminating closed basins (which I understand to be the problem that elicited the *ad hoc* approach), there are plenty of basin filling algorithms that would also solve this problem in a somewhat more rigorous way.

This consideredation should also be included when using the SIA to infer thickness fields from the balance flux. Contemporary surface DEMs often show topography at a much smaller scale than that which is relevant for determining driving stress in the Stokes' equations, and smoothing is necessary to avoid non-physical oscillations.

**1.5 $\dot{a}$ as a control variable**

The $\dot{a}$ resulting from the inversion procedure needs to be presented, because the derived thickness field only conserves mass with respect to this augmented field, and in my experience performing these types of inversions for $\dot{a}$ without considerable explicit smoothing leads to an apparent mass balance that can look pretty weird. Indeed, it becomes the dumping ground for all manner of errors derived from other input fields and the model itself. One way of getting around this is to apply a regularization term to $\dot{a}$ (with a regularization parameter associated with the length scale of feasible spatial variability in apparent mass balance), just as it is already applied to $H$ in Eq. 4.

**1.6 Boundary conditions**

The PDE being solved here (Eq. 1) involves only first spatial derivatives and no time derivative. As such, the number of boundary conditions allowed is limited to one per characteristic. An ice divide implicitly acts as a zero-flux boundary, meaning that specifying the flux at the margin is not well-defined. Nonetheless, the magic of numerical solutions allows it anyways. However, one needs to be quite careful to understand that

this introduces a fictitious source term (as it must for mass to be conserved) that needs to be accounted for in error estimates and interpretation of results. This artifact is evident in Fig. 4, along the eastern margin of VIC, where thickness is around 300m right up until it reaches a land terminating boundary. In a sense, this is the opposite problem of that which you're trying to solve with the first term in Eq. 4, but rather than the flux running out before reaching the margin, it doesn't go to zero fast enough. Perhaps consider introducing another term to Eq. 4 that adjust the surface mass balance so that flux goes to zero along land terminating boundaries?

**1.7 Formal error estimate**

This section is mostly based on Morlighem et al. (2010), with the key difference that flux at measurement locations is directly imposed via Dirichlet boundary conditions rather than a best match found through an inverse procedure. This is problematic, for the same reason discussed in the above point about boundary conditions: imposing the value of the flux at more than one location along a flightline must lead to a fictitious numerical source in order for the condition to be upheld, because the PDE is first-order and only admits a single boundary condition per characteristic. Thus there is a hidden source term that needs to be included in the estimate of $\dot{a}$ that is likely to locally exceed the (already very optimistic) error estimate of $0.2$m yr$^{-1}$.

On a related note, I think that the estimate of uncertainty in surface mass balance is probably incorrect following the inversion procedure. Consider the following line of reasoning: the PDE for error propagation is derived by stating that

$$\nabla \cdot (\mathbf{n} + \delta\mathbf{n})(F + \delta F) = \dot{a} + \delta\dot{a}, \tag{1}$$

which is separated into

$$(\nabla \cdot \mathbf{n}F - \dot{a}) + (\nabla \cdot [\mathbf{n}\delta F + \delta\mathbf{n}F] - \delta\dot{a}) = 0, \tag{2}$$

where $\mathbf{n}$ is the true (error free) flow direction, F the true flux, and $\dot{a}$ the true apparent mass balance, and the $\delta$-annotated quantities are the errors associated with each of these quantities. The first term is zero due to mass conservation, and the second is solved for $\delta F$ to get the error in the flux. The problem arises in the definition of $\delta\dot{a}$. In the manuscript it is given a numerical value (0.2m yr$^{-1}$), which is supposed to represent the observational uncertainty. However, after the optimization procedure is complete, which makes significant modifications to $\dot{a}$ (I assume, which is why it needs to be reported), this variable is no longer the value for which that particular error estimate holds. Instead, it is a new and potentially non-physical field into which has been placed error in surface elevation, smoothing lengths, numerical error, etc. The field resulting from the optimization is neither the true value $\dot{a}$, nor the original field for which the error estimate was made, and as such $\delta\dot{a}$ must pick up the remaining difference. I presume that it would thus be substantially larger than the initial 0.2m yr$^{-1}$ estimate and would be neither independent nor identically distributed.

As a final objection, I do not understand this business of taking the minimum of two error propagation PDEs, one with a reversed velocity field. Why should a more favorable error estimate propagate back upstream from an observation, given that hyperbolic PDEs transport information in one direction only (with respect to a characteristic). It is interesting to note that neither publication that initially stated this method (Morlighem et al., 2010, 2014) gives a reference for it, and I have never been able to find one in my own literature searches. I invite the authors to take this opportunity to convince me of this idea's correctness.

Now, lest I appear too curmudgeonly on this issue: is all this a big problem? Probably not. My Bayesian treatment of the problem (Brinkerhoff et al., 2016) suggested that the error estimate suggested here isn't too bad in a practical sense, and the cross-validation presented in the latter parts of this manuscript suggest the same. Additionally, the additive model of errors used in the PDE error propagation equations would tend to overestimate uncertainty (for the same reason that adding two Gaussian random variables doesn't double their standard deviation). However, I would like to see a more robust defense of the theory behind the methods used, and a more transparent accounting of the simplifying assumptions.

**1.8 Error estimate results**

The formal error estimate should provide an upper bound on the actual mismatch, not a prediction of it. A good metric here would be to compute the frequency by which the actual mismatch falls below the predicted error. In the context of normal distributions, the actual misfit would be less than the predicted misfit 95% (or whatever your definition of error is) of the time. This is more or less the definition of credibility intervals in Bayesian statistics. If the mismatch falls outside the estimated error much more frequently than this, then one begins to question what use the estimated error is. While it's sort of interesting to consider the median values, this neglects the aspect of error prediction which is likely to be of most interest to people who would use this product: spatial skill.

**1.9 Appendix A**

If one cannot attribute the observed flux to deformation in a physically reasonable way (e.g. with a material parameter $A$ that is physically justifiable), then doesn't it make sense to assume a certain amount of sliding? It seems to me that instead of allowing viscosity to be an order of magnitude less than temperate ice, one could adjust $\beta^2$ instead. This would make for a much more straightforward way of interpreting a figure like Fig. A1. I recognize that the authors may not wish to add the additional uncertainty of selecting a sliding law to the model, and I'm certainly fine with that. However, the notion that the viscosity parameter is aliasing unmodelled basal processes should at least be addressed.

**2 Minor Points and Technical Corrections**

NOTE: This paper has a relatively high number of typos and grammatical errors. I will try to point them out where I see them, but the manuscript would benefit considerably from detailed copy editing.

**P1L4** Please define what 'performs well' means.

**P1L12** 'Withholding parts': the paper shows this in a median sense, not a spatially explicit one, which is an important point.

**P1L16** 'are in fact'→ 'is'.

**P2L4** Delete 'large' (and non-specific adverbs throughout the manuscript at large).

**P2L5** 'thickness of the ice cover' → 'ice thickness'.

**P2L11** 'Antarctica Ice Sheet' → 'Antarctic Ice Sheet'.

**P2L13** 'thicknesses' → 'thickness'.

**P2L35** Perhaps elaborate on what 'computationally less favorable' means.

**P3L1** delete 'physical'.

**P3L7** I don't understand the implication of this sentence.

**P3L14** 'allows to estimate'→ 'allow estimation of'.

**P3L15** Missing period.

**P3L15** 'Much development ...' needs citation.

**P4L1** 'For DEMs and elevation changes...' this sentence doesn't really say anything.

**P4L1–8** Citations are needed throughout.

**Introduction at large** I suggest including a paragraph on the general availability of thickness observations for consistency.

**P4L23–24** This sentence needs a bit more specificity.

**General characteristics at large** This section should be compressed a bit to ensure that the information presented is relevant to the conclusions of the paper.

**Glacier outlines** why not use the modern high-res DEM everywhere?

**P6L11** 'For VIC, thickness measurements' → 'VIC thickness measurements'.

**P6L14** Delete 'there only'.

**P6L23** Do borehole depths agree with GPR?

**Figure 2** Is $\partial_t h <$ SMB along ice divides? This was a problem I encountered in ITMIX.

**P9L25** *Olex* needs a citation.

**P10L8** I think you can delete everything up to 'incompressibility can be written as ...'.

**All equations** The divergence operator is traditionally written as $\nabla\cdot$, rather than just $\nabla$, which is typically thought to mean the gradient operator.

**P10L28** 'Inflow boundaries': I don't understand this sentence.

**P11L18** I'm not sure I understand how using the spatial gradient in SMB solves the problem. Assuming that SMB is only a function of elevation, doesn't SMB also reach a maximum where slope goes to zero, i.e. SMB also has a very small slope where surface elevation does?

**P11L24** Not sure what is meant by 'ice-flux direction is positive'.

**Eq. 4** I think that the integral should be

$$\int_F^\infty \delta(s)\mathrm{d}s\mathrm{d}\Omega. \tag{3}$$

This means that if $F < 0$, then the integral crosses the origin, and the $\delta$ function is activated. As it is, the function penalizes positive flux values. Writing that integral as a Heaviside function might be more clear.

**P11L30** Maybe just state the parameter values and reasoning behind them, and forego subjective descriptions like 'good performance'.

**P12L11** In glaciology, aspect ratios under which the SIA applies are usually referred to as small, rather than large. This derives from the aspect ratio being the small paramter in the asymptotic analysis of the SIA.

**P12L24** Again, read Kamb and Echelmeyer (1986) for some theoretic basis for how to smooth for SIA applications.

**P12L29** A clearer sentence might read 'We apply a correction to the computed flux in order to avoid negative thickness values'.

**P13L2** Would it be possible to explain why these particular functions and parameters were chosen?

**P13L4** 'If no observations ...' I don't understand this sentence.

**Formal error estimate** There should be at least a mention of additional error induced by using the SIA. This should include error in the surface gradient norm, among other things.

**P14L27** 'to'→ 'too'.

**P115L1** Maybe make a reference to L-curve analysis here.

**P16L12** 'geoemetry' SIC.

**P16L17** Maybe note that for a region bounded by two flowlines, flux is always at a maximum at the ELA.

**P16L24–26** Is it that the old routing is still dominant on the surface topography, or that the velocity field and DEM aren't contemporaneous?

**P17L1** This line makes a strong case that $\delta\dot{a} = 0.2$ everywhere might not be right.

**Ice thickness and bedrock elevation at large** Is it possible to extend the results of your error analysis to integrated quantities like total sub-sea level area or total volume? It would make these numbers considerably more interesting if we could be sure that they represented a (statistically) significant departure from previous estimates.

**P22L26** 'Therefore ...' I don't understand this sentence.

**Figure 8** A gradient, rather than random colorbar would be useful here. Also, linear axes would help to get a sense for the sizes of the error bars.

**P24L4** 'none'→ 'any'.

**P25L23** Delete 'anyhow'.

**Ice thickness at large** Much of this section focuses on the differences between the first and second-step solution. Perhaps a figure showing the differences between the two predicted thickness fields would help the reader get a sense of how the differences are distributed?

**Figure 9** The transparency method for delineating which areas were subject to the second stage isn't very clear. Maybe switch colormaps or just draw a line around the areas that were updated.

**Error estimates at large** If error estimates go up when using the second stage, please convince me why the second stage is useful. Perhaps a similar plot to Fig. 8 is in order, which would hopefully show that including velocity observations reduces the actual mismatch for withheld measurements.

**P27L11** There's no 'might' about it: ignoring sliding biases the result towards thicker ice.

**P28L14** Perhaps consider reading Brinkerhoff (2016), which discusses this point in somewhat more detail.

**P29L23** 'mere' → 'a mere'.

**P30L6** 'tend to overestimate mismatch values' when taken in aggregate, but not necessarily individually.

**P30L7** 'Error estimates can here be considered upper and lower constraints of inferred thickeness values' → is this not the operational definition of error? If we knew that error was the exact amount by which our estimates were off, then we could just subtract it and get perfect results.

**P33L17** 'For glaciers ...' this sentence is pretty awkward.

**Figure A4** This might be better served by displaying a map of the difference in thickness between the two experiments.

**References**

Brinkerhoff, D. J., Aschwanden, A., and Truffer, M. (2016). Bayesian inference of subglacial topography using mass conservation. *Frontiers in Earth Science*, 4(8).

Farinotti, D., Brinkerhoff, D., Clarke, G. K. C., Fürst, J. J., Frey, H., Gantayat, P., Gillet-Chaulet, F., Girard, C., Huss, M., Leclercq, P. W., Linsbauer, A., Machguth, H., Martin, C., Maussion, F., Morlighem, M., Mosbeux, C., Pandit, A., Portmann, A., Rabatel, A., Ramsankaran, R., Reerink, T. J., Sanchez, O., Stentoft, P. A., Singh Kumari, S., van Pelt, W. J. J., Anderson, B., Benham, T., Binder, D., Dowdeswell, J. A., Fischer, A., Helfricht, K., Kutuzov, S., Lavrentiev, I., McNabb, R., Gudmundsson, G. H., Li, H., and Andreassen, L. M. (2016). How accurate are estimates of glacier ice thickness? results from itmix, the ice thickness models intercomparison experiment. *The Cryosphere Discussions*, 2016:1–34.

Kamb, B. and Echelmeyer, K. (1986). Stress-gradient coupling in glacier flow: I. longitudinal averaging of the influence of ice thickness and surface slope. *Journal of Glaciology*, 32(111).

Morlighem, M., Rignot, E., Mouginot, J., Seroussi, H., and Larour, E. (2014). High-resolution ice-thickness mapping in south greenland. *Annals of Glaciology*, 55(67).

Morlighem, M., Rignot, E., Seroussi, H., Larour, E., Ben Dhia, H., and Aubry, D. (2010). Spatial patterns of basal drag inferred using control methods from a full-stokes and simpler models for pine island glacier, west antarctica. *Geophysical Research Letters*, 37(14).
* * *

---

## Referee Comment (RC2) · F. Maussion (Referee) · 13 Apr 2017

**General comments**

In their manuscript, the authors present a new method to estimate the ice thickness of glaciers and ice-caps. They rely on an established theoretical background, but the paper presents innovative ways to deal with limited or inconsistent data input. The study is solid, comprehensive, and I am confident that many readers will find the paper useful for their research.

I agree with most (if not all) of the issues raised by D. Brinkerhoff, and there is no point

in repeating them here. However, I will modestly try to offer a different perspective, driven by my personal interests (large scale glaciology and reproducible science).

Generalisation of the method to other glaciers/regions

The authors state two times (in the abstract and in the conclusion) that their method has "*data requirements which are comparable to other approaches that have already been applied world-wide*". I have to disagree with this statement, which unnecessarily raises the reader's expectations. To my knowledge, we are still far away from a global dataset of surface mass-balance and $\partial h/\partial t$. The most promising method to estimate geodetic mass-balance (DEM differencing) is rarely applied to regions larger than a catchment or mountain chain, and the global methods (GRACE, Icesat) suffer from considerable drawbacks (coarse spatial resolution and high uncertainties). Therefore, I suggest to remove this statement from the abstract.

My understanding of the study is that it presents a way to deal with uncertainties in the boundary conditions and in the observations to which the model is tuned. Most efforts, it seems, are spent into correcting $\dot{a}$ to avoid singularities and in defining a formal way to propagate observational uncertainties. In the end, I feel like the paper would benefit from a more thorough discussion about the benefits and drawbacks of their method for large scale experiments, i.e. without any observation and/or without an observational dataset for $\dot{a}$.

Structure of the paper

Like D. Brinkerhoff, I notice that the paper could gain in readability. I am however unsure how to proceed. One the one hand, I truly appreciate the authors' thoroughness, and I am sure that the interested reader will find most of the information s/he needs to reproduce the steps listed here. On the other hand, the paper is long and sometimes difficult to follow. A change along the lines proposed by D. Brinkerhoff will surely improve the paper's readability, but I would refrain from cutting too much text out of the paper: instead, move some details to the appendix or the suppl. material. (take the paragraph about the slope angle threshold for example: a specialist will probably be interested in this information, but a more general audience would rather skip these details).

Case study?

To test a new method, one should rely on the best possible data input for calibration/validation. When looking at the fields in Fig. 02 I can't really imagine that this is the case in Svalbard. It is too late for this study, but in the future I would suggest to look at more appropriate benchmark glaciers (right at our doorstep?), where data denial experiments can be realised much more easily and with much more confidence in the boundary conditions.

**Specific comments**

**P2, L2, "virtually complete coverage"** none of the cited studies (apart maybe from Paul et al, which is rather a methodological review paper) states that surface elevation change products have reached complete coverage. This is related to my general comment: the method presented in this paper is promising, but still belongs to the "demanding ones" in terms of data availability.

**P2 L27, "apparent flux divergence"** I also think that this new terminology makes no sense. One can argue about whether "apparent mass-balance" is the best term or not, but "apparent flux divergence" is definitely more confusing than helping.

**Figure 5** Obviously, both the observations and the ice divides (zero flux locations) are way too visible on the bedrock topography. This calls for a more constrained tuning, either by changing the way $B$ is allowed to vary or by changing the way the model is dealing with small slopes?

**Figure 8** I understand the reason for using normalized values here, but from a volume estimation perspective (e.g. sea level rise), other metrics are much more important: bias and absolute error (i.e. small relative errors for large thickness values can be more relevant than large relative errors for small thickness values). Have you considered looking at absolute values, too?

**Editorial comments**

**P3 L15** dot is missing

**Fig. 02** intuitively, I associate blue values with positive mass-balance / surface change and red with the opposite. Consider reversing your colortable.

**All figures** consider damping the topographical shading, which is currently very strong without a clear added value.

**All figures** consider using another colormap. Rainbow (or "jet" in python) is now considered by many as being misleading (e.g. https://www.climate-lab-book.ac.uk/2014/end-of-the-rainbow/ and many further refs online)

---

## Author Comment (AC1) · 8 Jun 2017

O. Eisen
olaf.eisen@awi.de

*First of all we want to thank the reviewer for constructive comments on our manuscript. All comments have been taken into account and a list of answers and undertaken actions is given below. Answers are indented and in italic type while a **short summarising reply** is provided in italic, bold-face type.*

The authors spend some effort on providing formal estimates on the overall accuracy of the method, which is greatly appreciated. I suggest that they also look into the detailed comparison of various radar processing schemes provided by Moran et al. (2000). Those authors compare raw 1D vs. 2D vs 3D array radar data processing and come up with estimates of ice thickness errors which are larger than those often used in other studies. Lapazaran, as the most recent and comprehensive study on this issue, cite it, too, but the original paper might be worth looking into, as your study also includes very mountainous valleys.

> *We certainly appreciate this comment and checked how the error estimate in Moran et al. (2000) compares to the study from Lapazaran et al. (2016), which we considered. Moran et al. (2000) describe differences between inferred thickness values from treating the GPR data as single measurements, line measurements or as a 3D measurement array during the processing. They only give relative differences for their specific setup, which are difficult to transfer to our geometries. Lapazaran et al. (2016) recognise that this uncertainty term can become important. Yet they are unable to estimate it for Werenskioldbreen and were forced to discard it from their analysis. However, they gave an advise for how this term should be included if known.*

> *As our results are based on the same GPR measurements as used in Lapazaran et al. (2016), we follow their analysis and thus ignore this extra source term in the measurement error.*

> ***Added reference** to Moran et al. (2000) stating that the analysis of Lapazaran et al. (2016) discards an important but not well quantifiable source term in the error analysis.*

Delineation of a complexly dipping temperate glacier bed using short-pulse radar arrays Moran, M. L.; Greenfield, R. J.; Arcone, S. A.; Delaney, A. J. Journal of Glaciology, Volume 46, Number 153, March 2000, pp. 274-286(13)

---

## Author Comment (AC2) · 8 Jun 2017

D. Brinkerhoff (Referee #1)
douglas.brinkerhoff@gmail.com

*First of all we want to thank the reviewer for constructive comments on our manuscript. All comments have been taken into account and a list of answers and undertaken actions is given below. Answers are indented and in italic type while a **short summarising reply** is provided in italic, bold-face type.*

**Summary**
In this study, Fürst and others present an updated method for solving the problem of inferring ice thickness from sparse observations coupled with surface data. They then apply this method to three glacial systems on the Svalbard Archipelago. The validity of the resulting estimations of ice thickness are established with a detailed error analysis.

The paper is clearly relevant and within the appropriate scope for the journal. Contemporary interest in methods for inferring ice thickness are of great interest to the community, as evinced by numerous recent publications on the subject, including a comprehensive intercomparison (Farinotti et al., 2016). This paper's contribution to the field stems primarily from its presentation of a way to circumvent some of the arduous data requirements required by the method on which it is based (Morlighem et al., 2010). I am concerned that this paper inherits some of the potential shortcomings from that work, namely an error analysis which is not, in my view, completely justified, as well as a misunderstanding of the definition of error for PDE-constrained optimization. Nonetheless, the manuscript does a commendable job with respect to discussing its own limitations and in firmly placing the issue in the context of error analysis. I am not sure that the paper will provoke a sea-change in thinking about Svalbard's glaciological processes, but I imagine that the results will be useful for modellers and others needing ice thickness estimates.

Stylistically, I think that the paper could benefit from significant distillation. The elimination of superfluous words, sentences, and perhaps even sections would help the reader to focus on essential points. As it stands, the manuscript feels like a methods paper mixed up with a case study. A stronger partitioning between these two parts would help. The paper also contains a fair amount of questionable English. I have tried to make corrections where I can, but a more detailed reading by the authors themselves is in order.

*The manuscript was re-structured and copy-edited by native speakers during*

*the revision. In addition, an effort was made to further distill the presentation of the results.*

Ultimately, under the assumption that the authors can address the criticisms that I have presented below, I would encourage resubmission.

**1 Major Points**

**1.1 On the use of "apparent flux divergence"**

The phrase "apparent mass balance" is well ensconced in the literature at this point, and the reasoning for its use is fairly clear: when $\partial_t h$ is used as an observation, it acts identically to $\dot{b}$, which is a source term. Combining them leads to a simplified equation involving the flux divergence and this unified source term. The term 'apparent flux divergence' makes no sense at all. In fact, a more correct statement would be to just call the apparent mass balance the flux divergence (rather than apparent flux divergence), because that is what the equal sign implies. However, this would be confusing and tautological to say that the flux divergence equals the flux divergence. It is equally confusing, but perhaps less correct to say that the apparent flux divergence equals the flux divergence. If it's not clear already, I suggest using the term 'apparent mass balance' instead.

> *As the reviewer rightly points out, the 'apparent mass balance' is a difference between all mass balance terms (internal, surface and basal accumulation and ablation) and the surface elevation changes $\partial_t h$. Relying on the 'Glossary for glacier mass balance', this difference is certainly not referred to as a mass balance, except if $\partial_t h$ is zero, which is most evident. That's why we initially refrained from invoking the term 'mass balance'. Yet we understand the concern of the reviewer on our choice for the term 'apparent flux divergence'. In lack of a better alternative and since the term 'apparent mass balance' is somehow already established in the literature, we adjusted the manuscript appropriately.*
>
> **Corrected** *as suggested.*

**1.2 General characteristics section, and the extensive use of proper nouns**

This paper walks a difficult line between a methods paper and a case study. I don't have a problem with that, as it's generally useful to see methods applied to real cases to evaluate their worth. However, I think that the organization of this paper is such that it can be quite confusing. I would suggest reorganizing the paper such that the methods are completely stated (including the theory behind the error analysis; more on that in a minute), then switch gears and begin discussing the nuances of Svalbard's geometry and data availability. Thus I would not have to remember what and where Austre Torrellbreen is after reading many pages about hyperbolic PDEs.

*Agreed. We rearranged the manuscript accordingly and present the methodology prior to the presentation of the study site.*

***Corrected** as suggested.*

Additionally, all figures need to be labelled with salient features discussed in the text. The reader should not have to cross-reference Figure 2, while at the same time reading the paper text and analyzing the figure content. Additionally, some basic annotations describing some particular key points referenced in the text and restated in the figure captions would be very helpful.

*We now labelled the more salient features in most figures. Moreover, some particular key points were added in the captions.*

***Corrected** as suggested.*

1.3 Estimates of input observation error

Input data error estimates need to be stated more clearly and with more complete justification. For example, the authors use a mean value of 5 m for estimated thickness error for all experiments, based on a (convincing) paper in which GPR was applied to a relatively thin glacial system, with lower wavelength antennae. It is not tenable to assume the same error estimate for airborne radar measurement from the 1980s. Also, no real effort is made to justify the 0.2 m yr$^{-1}$ estimate in apparent surface mass balance.

*Concerning the measurement error associated with the used thickness data, we now decided to distinguish between GPR and airborne surveys on VIC. Petterrson et al. (2010) estimate the error for these surveys to be 9.3 and 23.1 m, respectively. For the formal error estimate on VIC, we now use 10 and 25 m. For WSB and THPB, we kept the 5-m estimate from Lapazaran et al. (2016), who relied on a WSB portion of the thickness measurements used in our manuscript.*

*Concerning the SMB uncertainty estimate, we rely on Lang et al. (2015) who compared their simulated SMB values to observations. This comparison gave the means to justify an uncertainty estimate. In the SMB input description, we now specify the following:*

'*The difference between modelled SMB values and 10 used validation sites shows a low bias of -0.03 m i.e. yr$^{-1}$ with a standard deviation of 0.14 m i.e. yr$^{-1}$. The latter value is considered as an uncertainty estimate for the SMB field.*'

*Concerning the error in $\partial_t h$, Moholdt et al. (2010) compute an average 0.3 m yr$^{-1}$ at all cross-over points for their 'plane method'. In the section on the $\partial_t h$ input, we now added:*

'*Moholdt et al. (2010) report that the local root-mean-square deviation of several hundred*

*surface-change estimates is 0.3 m yr⁻¹. ..'*

> **Corrected** *by adjusting the airborne and ground RES measurement uncertainty on VIC. Moreover, we now provide some motivation for a 0.4 m yr⁻¹ input uncertainty for the apparent mass balance.*

For all of these error sources, the distinction between simple observational error and the error associated with the variables used in your equations (which are time averaged) needs to be discussed. See the methods section of Brinkerhoff et al. (2016) for a discussion of what I mean by this; in short, Eq. 1 assumes that all of the inputs are temporally consistent, but in reality they are not, and this induces an additional source of uncertainty.

> *In reponse to this appreciated comment we added a passage to the description of the formal error estimate (new Sect. 2.2.3; former Sect. 3.2.3)*
>
> *'[...], the error analysis accounts for uncertainties in the observational record but does not comprise the error that stems from time-averaging of the input and temporal inconsistencies between different fields. A detailed assessment and treatment of various input errors within a Bayesian framework is presented in Brinkerhoff et al. (2016).'*
>
> **Corrected** *by adding a brief discussion on this error source term and pointing to the suggested reference.*

**1.4 Flow directions**

I strongly suggest reading Kamb and Echelmeyer (1986), when approaching flow direction computations. They provide a strong theoretical basis for how to approach ice routing using only surface elevation observations in order to approximately recover higher-order directions. With the availability of these theoretical results, which have been used successfully many times throughout the balance flux literature, I find the ad-hoc method of smoothing by applying a contouring algorithm then using natural neigh- bors interpolation to degrade the surface elevation accuracy to be troubling. Barring the success of the smoothing proposed by Kamb and Echelmeyer (1986) at eliminating closed basins (which I understand to be the problem that elicited the ad hoc approach), there are plenty of basin filling algorithms that would also solve this problem in a somewhat more rigorous way.

> *This comment of the reviewer is most appreciated because the suggestion to refer to the balance flux literature did substantiate the calculation of the flux direction. We now decided to follow the approach presented in Brinkerhoff & Johnson (2015) for the determination of the flux direction (Eqs. 9 and 10) by solving a partial differential equation for a smoothed version of the driving stress. In this way, the direction choice is based on theoretical arguments and it is very convenient that the smoothing radius varies with the ice*

*thickness and thus naturally adjusts to different glacier sizes.*

**Corrected** *by opting for the direction choice forwarded in Brinkerhoff &
Johnson (2015).*

This consideredation should also be included when using the SIA to infer thickness
fields from the balance flux. Contemporary surface DEMs often show topography
at a much smaller scale than that which is relevant for determining driving stress
in the Stokes' equations, and smoothing is necessary to avoid non-physical oscil-
lations.

*The smoothed driving stress solution is now also used in terms of a smooth
surface topography entering the SIA equation.*

**Corrected** *as suggested.*

1.5 $\dot{a}$ as a control variable

The $\dot{a}$ resulting from the inversion procedure needs to be presented, because the
derived thickness field only conserves mass with respect to this augmented field,
and in my experience performing these types of inversions for $\dot{a}$ without consid-
erable explicit smoothing leads to an apparent mass balance that can look pretty
weird. Indeed, it becomes the dumping ground for all manner of errors derived
from other input fields and the model itself. One way of getting around this is to
apply a regularization term to $\dot{a}$ (with a regularization parameter associated with
the length scale of feasible spatial variability in apparent mass balance), just as it
is already applied to $H$ in Eq. 4.

*Initially, we did not provide these fields because the article was already
rather long and the information gained from this comparison is of interest to
a rather limited community. Anyhow, differences are instructive and provide
some intuitive understanding on how the optimisation works. Therefore, we
provide now another figure showing the final apparent mass balance fields
for both regions and a brief discussion, both in a separate Appendix section.
The difference fields show no dominant spatial variability pattern, which
would justify another regularisation term in the cost function. Moreover, a
general bias/drift was already prevented by penalising differences between
the initial and the final apparent mass balance. Therefore, we refrained from
any modification of the cost function.*

**Corrected** *by adding the requested figure and a brief discussion in the Ap-
pendix. We did not see the necessity to add the suggested regularisation term
to the cost function.*

1.6 Boundary conditions

The PDE being solved here (Eq. 1) involves only first spatial derivatives and no

time derivative. As such, the number of boundary conditions allowed is limited to one per characteristic. An ice divide implicitly acts as a zero-flux boundary, meaning that specifying the flux at the margin is not well-defined. Nonetheless, the magic of numerical solutions allows it anyways. However, one needs to be quite careful to understand that this introduces a fictitious source term (as it must for mass to be conserved) that needs to be accounted for in error estimates and interpretation of results. This artifact is evident in Fig. 4, along the eastern margin of VIC, where thickness is around 300m right up until it reaches a land terminating boundary. In a sense, this is the opposite problem of that which you're trying to solve with the first term in Eq. 4, but rather than the flux running out before reaching the margin, it doesn't go to zero fast enough. Perhaps consider introducing another term to Eq. 4 that adjust the surface mass balance so that flux goes to zero along land terminating boundaries?

> *The reviewer rightly points out that there are fictitious source and sink terms in the ice flux at any boundaries. These terms accommodate inconsistencies particularly in areas where multiple Dirichlet conditions are set. These terms are not necessarily biased as values are both negative and positive for a single test geometry. We followed the reviewers suggestion and made several attempts to reduce these fictitious values by directly penalising them or by increasing the flux-solution smoothing around the land-terminated margin. The solution could actually be improved near the margin for some glaciers. Yet for others, a worthwhile reduction of the magnitude of these boundary source and sink terms could only be achieved at the expense of other terms in the cost. We could not find settings that resulted in an overall improvement for all test geometries. We therefore cannot offer a solution within this revision as more development is necessary.*
>
> ***Not corrected.*** *Several attempts to solve the raised issue did fail and require more development.*
>
> *The thickness artifact, the reviewer mentions (Fig. 4), is certainly associated with a flux field that does not necessarily decrease most gradually to zero as the margin is reached. Yet the mentioned area also shows that the surface elevation reaches 300m near the margin. We do not think that this is a real feature but rather some inconsistency between the chosen DEM and the glacier inventory margin position. Along other segments of the land-terminated margin, ice thickness decreases more gradually.*
>
> *Though not specifically asked here, we removed the Dirichlet condition on ice thickness at the measurements locations and replcaed it by another term in the cost function. In this way, further fictitious source and sink terms in the solution are avoided.*
>
> ***Replaced internal Dirichlet conditions*** *in second-step reconstruction by extra cost term.*

1.7 Formal error estimate

This section is mostly based on Morlighem et al. (2010), with the key difference that flux at measurement locations is directly imposed via Dirichlet boundary conditions rather than a best match found through an inverse procedure. This is problematic, for the same reason discussed in the above point about boundary conditions: imposing the value of the flux at more than one location along a flightline must lead to a fictitious numerical source in order for the condition to be upheld, because the PDE is first-order and only admits a single boundary condition per characteristic. Thus there is a hidden source term that needs to be included in the estimate of $\dot{a}$ that is likely to locally exceed the (already very optimistic) error estimate of 0.2m yr$^{-1}$.

On a related note, I think that the estimate of uncertainty in surface mass balance is probably incorrect following the inversion procedure. Consider the following line of reasoning: the PDE for error propagation is derived by stating that

$$\nabla \cdot (\mathbf{n} + \delta\mathbf{n})(F + \delta F) = \dot{a} + \delta\dot{a} \tag{1}$$

which is separated into

$$(\nabla \cdot \mathbf{n}F - \dot{a}) + (\nabla[\mathbf{n}F + \delta\mathbf{n}F] - \dot{a}) = 0, \tag{2}$$

where n is the true (error free) flow direction, F the true flux, and $\dot{a}$ the true apparent mass balance, and the -annotated quantities are the errors associated with each of these quantities. The first term is zero due to mass conservation, and the second is solved for $\delta F$ to get the error in the flux. The problem arises in the definition of $\delta\dot{a}$. In the manuscript it is given a numerical value (0.2m yr$^{-1}$), which is supposed to represent the observational uncertainty. However, after the optimization procedure is complete, which makes significant modifications to $\dot{a}$ (I assume, which is why it needs to be reported), this variable is no longer the value for which that particular error estimate holds. Instead, it is a new and potentially non-physical field into which has been placed error in surface elevation, smoothing lengths, numerical error, etc. The field resulting from the optimization is neither the true value $\dot{a}$, nor the original field for which the error estimate was made, and as such $\delta\dot{a}$ must pick up the remaining difference. I presume that it would thus be substantially larger than the initial 0.2m yr$^{-1}$ estimate and would be neither independent nor identically distributed.

As a final objection, I do not understand this business of taking the minimum of two error propagation PDEs, one with a reversed velocity field. Why should a more favorable error estimate propagate back upstream from an observation, given that hyperbolic PDEs transport information in one direction only (with respect to a characteristic). It is interesting to note that neither publication that initially stated this method (Morlighem et al., 2010, 2014) gives a reference for it, and I have never been able to find one in my own literature searches. I invite the authors to

take this opportunity to convince me of this idea's correctness.

Now, lest I appear too curmudgeonly on this issue: is all this a big problem? Probably not. My Bayesian treatment of the problem (Brinkerhoff et al., 2016) suggested that the error estimate suggested here isn't too bad in a practical sense, and the cross-validation presented in the latter parts of this manuscript suggest the same. Additionally, the additive model of errors used in the PDE error propagation equations would tend to overestimate uncertainty (for the same reason that adding two Gaussian random variables doesn't double their standard deviation). However, I would like to see a more robust defense of the theory behind the methods used, and a more transparent accounting of the simplifying assumptions.

*The reviewer is rather critical about the presented error estimation, which was certainly never intended to be a strict probabilistic measure. Yet, the presented error estimation is of a practical use and it should be able to distinguish areas in which the thickness reconstruction is well constrained from others where this is not the case. In the most practical sense, the latter areas could be the target for future survey campaigns. We certainly do not want to promote the error estimate field as an input for further error analysis. Here, our concern is that the field could be picked up for assessing the sensitivity of glacier volume projections to bedrock changes. For such a purpose, more stringent statistical measures should be invoked and implemented. One of the next development steps certainly includes the implementation of a more stringent treatment of uncertainties from diverse sources.*

*The reviewer also asked for a more robust defense of the error-estimate theory and its limitations/assumptions. This is not evident because our approach is informed by Morlighem et al. (2011), who did not provide any reference. Anyhow, we gladly try to provide more information on assumption and limitations. In this approach, we have a certain control on the error at the measurement location from the acquisition instruments/settings. The central assumption is then of course that the erroneous flux field also fulfills the mass conservation, which implies that the error is propagated along the flow downglacier. This assumption might not be too problematic as long as input uncertainties do not dominate. Exclusive downstream error propagation results in a sawtooth structure with error estimate that increase gradually until a next downstream measurement is reaches where values drop abruptly. To avoid this structure, we followed the upstream propagation presented in Morlighem et al. (2011). A first argument why this might anyhow hold comes from the mass conservation equation itself. A multiplication by (-1.0) is analytically identical though numerically different. In addition, the physical interpretation is upward motion with an inverted source and sink field. We however understand the concern that the error should not propagate upstream against the ice flux. The reviewer might however agree that*

*a measurement constrains the thickness field also some way upstream. This would imply that the error has to decrease along a flowline the closer one gets to a measurement. The rate of this error decrease is certainly questionable. If we accept that errors are allowed to decrease with a certain rate along the flow, then upstream Dirichlet boundary conditions have to be set such that the error equals the measurement uncertainty at the survey locations (further downstream). To improve the conditioning of the problem, we multiply by (-1.0), which is physically interpreted as upstream propagation. Yet, there is a slight nuance in this because we have to deliberately decide now for a rate of error change valid for upstream propagation. This rate might differ from the downstream propagation but as we have no good reason for choosing a different value, we simply set magnitudes equal from the downstream propagation. A similar discussion is added to the text and the reader will be prominently informed about the limitations of upstream error propagation, using a disputable change rate.*

**Expanded discussion** *on the assumptions in the formulation of upstream error propagation.*

*The reviewer also raises concerns that the modification of the apparent mass balance field during the optimisation will affect the input uncertainty distribution (assumed constant). Even though a spatially variable input uncertainty field could be accommodated, it is far from evident how to generate such a field during the iterative update of the apparent mass balance within our rather basic error-estimation approach. A more stringent treatment of error distribution would be required, which we like to postpone to further development of our approach. Anyhow, we added a brief passage that the assumption of a constant input uncertainty of the apparent mass balance (and likewise for the surface velocities in the second step) might partially be undermined/corrupted or limited during the iterative update of the control parameters. In addition, the relevance of this concern can be assessed from the now provided comparison between the initial and the final apparent mass balance fields in the Appendix.*

**Added a brief passage on this particular concern** *in the manuscript. Necessary development work for a stringent treatment of various error source terms (as for instance in Bayesian statistics), though not directly requested by the reviewer, goes beyond this revision and would justify another submission.*

1.8 Error estimate results

The formal error estimate should provide an upper bound on the actual mismatch, not a prediction of it. A good metric here would be to compute the frequency by which the actual mismatch falls below the predicted error. In the context of normal distributions, the actual misfit would be less than the predicted misfit 95% (or

whatever your definition of error is) of the time. This is more or less the definition of credibility intervals in Bayesian statistics. If the mismatch falls outside the estimated error much more frequently than this, then one begins to question what use the estimated error is. While it's sort of interesting to consider the median values, this neglects the aspect of error prediction which is likely to be of most interest to people who would use this product: spatial skill.

> *Again the reviewer had a good point on the spatial skill of the error-estimate maps. We therefore added all locations where the min/max interpretation of the error estimates is violated to Fig. 9 (now Fig. 8). Moreover a table is added giving the fraction of the withheld measurements for which the actual mismatch is smaller than the formal error estimate. We find that the min/max interpretation valid in more than 80% of the cases, except if the reconstruction is informed by very few measurments.*

> **Corrected** *by adding a Table with the requested information.*

1.9 Appendix A

If one cannot attribute the observed flux to deformation in a physically reasonable way (e.g. with a material parameter A that is physically justifiable), then doesn't it make sense to assume a certain amount of sliding? It seems to me that instead of allowing viscosity to be an order of magnitude less than temperate ice, one could adjust $\beta^2$ instead. This would make for a much more straightforward way of interpreting a figure like Fig. A1. I recognize that the authors may not wish to add the additional uncertainty of selecting a sliding law to the model, and I'm certainly fine with that. However, the notion that the viscosity parameter is aliasing unmodelled basal processes should at least be addressed.

> *We refrained from choosing an additional sliding law but rather invoke the notion that the viscosity parameter is aliased by unaccounted model physics.*

> **Added** *notion on unaccounted physics aliasing the tuning of the viscosity parameter.*

**2 Minor Points and Technical Corrections**

NOTE: This paper has a relatively high number of typos and grammatical errors. I will try to point them out where I see them, but the manuscript would benefit considerably from detailed copy editing.

**P1L4** Please define what 'performs well' means.

> **Reformulated** *as follows.*

> *'The approach is applied to a variety of test geometries [...]'*

**P1L12** 'Withholding parts': the paper shows this in a median sense, not a spatially explicit one, which is an important point.

> ***Corrected*** *by adding the adjective 'median'.*

**P1L16** 'are in fact'⟶ 'is'.

> ***Corrected*** *as suggested.*

**P2L4** Delete 'large' (and non-specific adverbs throughout the manuscript at large).

> ***Done*** *during the copy-edit.*

**P2L5** 'thickness of the ice cover' ⟶ 'ice thickness'.

> ***Corrected*** *as suggested.*

**P2L11** 'Antarctica Ice Sheet' ⟶ 'Antarctic Ice Sheet'.

> ***Corrected*** *as suggested.*

**P2L13** 'thicknesses' ⟶ 'thickness'.

> ***Corrected*** *as suggested throughout the manuscript.*

**P2L35** Perhaps elaborate on what 'computationally less favorable' means.

> ***Reformulated.*** *'[...] at the expense of computational costs.'*

**P3L1** delete 'physical'.

> ***Done***.

**P3L7** I don't understand the implication of this sentence.

> ***Deleted sentence***.

**P3L14** 'allows to estimate'⟶ 'allow estimation of'.

> ***Done***.

**P3L15** Missing period.

> ***Inserted*** *full stop.*

**P3L15** 'Much development ...' needs citation.

> ***Inserted*** *relevant reference.*

**P4L1** 'For DEMs and elevation changes...' this sentence doesn't really say anything.

> ***Deleted sentence***.

**P4L18** Citations are needed throughout.

> ***Added relevant citations*** *for remote-sensing based DEM generation and velocity measurements.*

**Introduction at large** I suggest including a paragraph on the general availability of thickness observations for consistency.

> ***Added*** *two sentences on the availability of thickness measurements referring to GlaThiDa 2.0.*

**P4L2324** This sentence needs a bit more specificity.

> ***Deleted sentence.***

**General characteristics at large** This section should be compressed a bit to ensure that the information presented is relevant to the conclusions of the paper.

> ***Paragraph was shortened.***

**Glacier outlines** why not use the modern high-res DEM everywhere?

> ***No action taken.*** *High-resolution DEM is not available for Southern Spitsbergen.*

**P6L11** 'For VIC, thickness measurements' $\longrightarrow$ 'VIC thickness measurements'.

> ***Corrected*** *as suggested.*

**P6L14** Delete 'there only'.

> ***Done.***

**P6L23** Do borehole depths agree with GPR?

> ***Values agree on view.*** *A direct comparison is difficult as the positioning of these icecores has an uncertainty of roughly 100m. For the deep borehole B closest to the ice front (bed contact after 206m; Jania et al., 1996), the GPR survey line shows a local thickness maximum of about 200m.*

**Figure 2** Is $\partial_t h <$ SMB along ice divides? This was a problem I encountered in IT-MIX.

> ***No action necessary.*** *Panels (e) and (f) of Fig.2 show that the apparent mass balance is positive over most of the divide areas.*

**P9L25** Olex needs a citation.

> ***Removed sentence*** *because a co-author also commented that the Olex system is mentioned too prominently.*

**P10L8** I think you can delete everything up to 'incompressibility can be written as ...'.

> ***Corrected*** *as suggested.*

**All equations** The divergence operator is traditionally written as $\nabla\cdot$, rather than just $\nabla$, which is typically thought to mean the gradient operator.

***Corrected** as suggested.*

**P10L28** 'Inflow boundaries': I don't understand this sentence.

> ***Reformulated sentence** as follows.*
>
> *'Inflow boundaries did not occur in our setup. These would require Dirichlet conditions on the ice flux.'*

**P11L18** I'm not sure I understand how using the spatial gradient in SMB solves the problem. Assuming that SMB is only a function of elevation, doesn't SMB also reach a maximum where slope goes to zero, i.e. SMB also has a very small slope where surface elevation does?

> ***Sentence was removed** as SMB gradients are in fact only used where surface slopes become exactly zero. Initially, it was wrongly stated that the SMB gradients replace the surface slopes where their magnitude is less than $\alpha_0$. Actually this emergency strategy is only necessary for some marine glacier snouts for which the DEM is completely flat for instance where the DEM and the glacier inventory are inconsistent). The SMB calculations rely on a different geometry and were therefore free of these artifacts.*

**P11L24** Not sure what is meant by 'ice-flux direction is positive'.

> ***Corrected** to 'flux magnitude is positive'.*

**Eq. 4** I think that the integral should be

$$\int_F^\infty \delta(s)\mathrm{d}s\mathrm{d}\Omega \tag{3}$$

This means that if $F < 0$, then the integral crosses the origin, and the $\delta$ function is activated. As it is, the function penalizes positive flux values. Writing that integral as a Heaviside function might be more clear.

> ***Corrected** by rewriting the integral as a Heaviside function.*

**P11L30** Maybe just state the parameter values and reasoning behind them, and forego subjective descriptions like 'good performance'.

> ***Removed** the subjective description.*

**P12L11** In glaciology, aspect ratios under which the SIA applies are usually referred to as small, rather than large. This derives from the aspect ratio being the small paramter in the asymptotic analysis of the SIA.

> ***Corrected**.*

**P12L24** Again, read Kamb and Echelmeyer (1986) for some theoretic basis for how to smooth for SIA applications.

> *Corrected. We now consistently use a smoothed surface-slope field following Brinkerhoff and Johnson (2015).*

**P12L29** A clearer sentence might read 'We apply a correction to the computed flux in order to avoid negative thickness values'.

> *Corrected as suggested.*

**P13L2** Would it be possible to explain why these particular functions and parameters were chosen?

> *The details of the flux correction is now presented in the Appendix. In this way, the general reader is not interrupted by the technicalities of this correction. The appendix now allows more space for explaining the choice of the functional dependence and the parameter choice. **Paragraph was moved to the Appendix** where more specifications are given.*

**P13L4** 'If no observations ...' I don't understand this sentence.

> *Reformulated the passage accordingly.*

**Formal error estimate** There should be at least a mention of additional error induced by using the SIA. This should include error in the surface gradient norm, among other things.

> ***Added** not on these error sources.*

**P14L27** 'to' ⟶ 'too'.

> ***Done.***

**P115L1** Maybe make a reference to L-curve analysis here.

> ***No action undertaken.** We refrained from a strict L-curve analysis because it does neither give an objective criterion for parameter selection. The analysis will not be straight forward as 5 multipliers have to be considered.*

**P16L12** 'geoemetry' SIC.

> ***Done.***

**P16L17** Maybe note that for a region bounded by two flowlines, flux is always at a maximum at the ELA.

> ***No correction necessary.** We are not sure if this generally holds when $\partial_t h \neq 0$.*

**P16L24-26** Is it that the old routing is still dominant on the surface topography, or that the velocity field and DEM aren't contemporaneous?

> *Reformulated paragraph. The VIC surface geometry is from 2010 and our ice velocities from 2015-16 so both explanation hold. Yet 2008 velocities already show that both branches were active (Pohjola et al., 2011). Therefore, the reason is that the old routing is still dominantly imprinted on the topography.*

**P17L1** This line makes a strong case that $\delta\dot{a} = 0.2$ everywhere might not be right.

> *We increased the input uncertainty $\delta\dot{a}$.*

**Ice thickness and bedrock elevation at large** Is it possible to extend the results of your error analysis to integrated quantities like total sub-sea level area or total volume? It would make these numbers considerably more interesting if we could be sure that they represented a (statistically) significant departure from previous estimates.

> *The interpretation of median error-estimates as upper constraints on the aggregate mismatch can certainly be transferred to the ice volume and the area fraction below sea-level. Even if error estimates are rather high, the previous estimate for ice grounded below sea-level on VIC of 5% falls outside the maximum range of 6 - 23%. We also provide now maximum ranges for the mean ice thickness. These ranges are given for both the first and second step in the reconstruction approach.*
>
> *Added maximum ranges for the integrated quantities.*

**P22L26** 'Therefore ...' I don't understand this sentence.

> ***Removed sentence.***

**Figure 8** A gradient, rather than random colorbar would be useful here. Also, linear axes would help to get a sense for the sizes of the error bars.

> ***No action undertaken.** In the presented way, colours are clearly distinguishable. ColorBrewer does not advice the usage of a sequential or divergent (both gradual) colour map with 12 colours. Results cluster and overlap using linear axes .*

**P24L4** 'none' $\longrightarrow$ 'any'.

> ***Done.***

**P25L23** Delete 'anyhow'.

> ***Done.***

**Ice thickness at large** Much of this section focuses on the differences between the first and second-step solution. Perhaps a figure showing the differences between the two predicted thickness fields would help the reader get a sense of how the differences are distributed?

*We refrain from adding a difference plot because its interpretation is not necessarily more evident than comparing absolute values.*

***No action undertaken***.

**Figure 9** The transparency method for delineating which areas were subject to the second stage isn't very clear. Maybe switch colormaps or just draw a line around the areas that were updated. Error estimates at large If error estimates go up when using the second stage, please convince me why the second stage is useful. Perhaps a similar plot to Fig. 8 is in order, which would hopefully show that including velocity observations reduces the actual mismatch for withheld measurements.

***Removed transparency and added median error-estimate/mismatch plot*** *similar to Fig. 8 (now Fig. 7).*

**P27L11** There's no 'might' about it: ignoring sliding biases the result towards thicker ice.

***Adjusted*** *as suggested.*

**P28L14** Perhaps consider reading Brinkerhoff (2016), which discusses this point in somewhat more detail.

***See above*** *corrections.*

**P29L23** 'mere' ⟶ 'a mere'.

***Done***.

**P30L6** 'tend to overestimate mismatch values' when taken in aggregate, but not necessarily individually.

***Corrected*** *by distinguishing between error estimates in terms of an aggregate median value and for individual measurements.*

**P30L7** 'Error estimates can here be considered upper and lower constraints of inferred thickeness values' ⟶ is this not the operational definition of error? If we knew that error was the exact amount by which our estimates were off, then we could just subtract it and get perfect results.

***Removed sentence***.

**P33L17** 'For glaciers ...' this sentence is pretty awkward.

***Removed sentence*** *because it held self-evident content.*

**Figure A4** This might be better served by displaying a map of the difference in thickness between the two experiments.

***Not adjusted*** *because zero thickness values are well visible in this map. Difference plots showed no improvement.*

**References**

Brinkerhoff, D. J., Aschwanden, A., and Truffer, M. (2016). Bayesian inference of subglacial topography using mass conservation. Frontiers in Earth Science, 4(8).

Farinotti, D., Brinkerhoff, D., Clarke, G. K. C., Fürst, J. J., Frey, H., Gantayat, P., Gillet-Chaulet, F., Girard, C., Huss, M., Leclercq, P. W., Linsbauer, A., Machguth, H., Martin, C., Maussion, F., Morlighem, M., Mosbeux, C., Pandit, A., Portmann, A., Rabatel, A., Ramsankaran, R., Reerink, T. J., Sanchez, O., Stentoft, P. A., Singh Kumari, S., van Pelt, W. J. J., Anderson, B., Benham, T., Binder, D., Dowdeswell, J. A., Fischer, A., Helfricht, K., Kutuzov, S., Lavrentiev, I., McNabb, R., Gudmundsson, G. H., Li, H., and Andreassen, L. M. (2016). How accurate are estimates of glacier ice thickness? results from itmix, the ice thickness models intercomparison experiment. The Cryosphere Discussions, 2016,1-34.

Kamb, B. and Echelmeyer, K. (1986). Stress-gradient coupling in glacier flow: I. longitudinal averaging of the influence of ice thickness and surface slope. Journal of Glaciology, 32(111).

Morlighem, M., Rignot, E., Mouginot, J., Seroussi, H., and Larour, E. (2014). High-resolution ice-thickness mapping in south greenland. Annals of Glaciology, 55(67).

Morlighem, M., Rignot, E., Seroussi, H., Larour, E., Ben Dhia, H., and Aubry, D. (2010). Spatial patterns of basal drag inferred using control methods from a full-stokes and simpler models for pine island glacier, west antarctica. Geophysical Research Letters, 37(14).

---

## Author Comment (AC3) · 8 Jun 2017

F. Maussion (Referee #2)
fabien.maussion@uibk.ac.at

*First of all we want to thank the reviewer for constructive comments on our manuscript. All comments have been taken into account and a list of answers and undertaken actions is given below. Answers are indented and in italic type while a* **short summarising reply** *is provided in italic, bold-face type.*

**General comments**

In their manuscript, the authors present a new method to estimate the ice thickness of glaciers and ice-caps. They rely on an established theoretical background, but the paper presents innovative ways to deal with limited or inconsistent data input. The study is solid, comprehensive, and I am confident that many readers will find the paper useful for their research. I agree with most (if not all) of the issues raised by D. Brinkerhoff, and there is no point in repeating them here. However, I will modestly try to offer a different perspective, driven by my personal interests (large scale glaciology and reproducible science).

Generalisation of the method to other glaciers/regions

The authors state two times (in the abstract and in the conclusion) that their method has "data requirements which are comparable to other approaches that have already been applied world-wide". I have to disagree with this statement, which unnecessarily raises the readers expectations. To my knowledge, we are still far away from a global dataset of surface mass-balance and $\partial h/\partial t$. The most promising method to estimate geodetic mass-balance (DEM differencing) is rarely applied to regions larger than a catchment or mountain chain, and the global methods (GRACE, Icesat) suffer from considerable drawbacks (coarse spatial resolution and high uncertainties). Therefore, I suggest to remove this statement from the abstract. My understanding of the study is that it presents a way to deal with uncertainties in the boundary conditions and in the observations to which the model is tuned. Most efforts, it seems, are spent into correcting $\dot{a}$ to avoid singularities and in defining a formal way to propagate observational uncertainties. In the end, I feel like the paper would benefit from a more thorough discussion about the benefits and drawbacks of their method for large scale experiments, i.e. without any observation and/or without an observational dataset for $\dot{a}$.

> *We agree with the reviewer that reconstruction approaches that are applied world-wide comprise an approximation for unknown input fields as SMB and*

*surface elevation changes. Such a parametersiation is indeed not presented here. So, our approach is not directly transferable worldwide. We therefore rephrased sentences in the abstract and the conclusion. In this way, no expectations on global applicability are raised. Therefore, a discussion on benefits and drawbacks of the presented approach in terms of global application become dispensable.*

***Corrected** by rephrasing sentences in the abstract and conclusion.*

Structure of the paper

Like D. Brinkerhoff, I notice that the paper could gain in readability. I am however unsure how to proceed. One the one hand, I truly appreciate the authors thoroughness, and I am sure that the interested reader will find most of the information s/he needs to reproduce the steps listed here. On the other hand, the paper is long and sometimes difficult to follow. A change along the lines proposed by D. Brinkerhoff will surely improve the papers readability, but I would refrain from cutting too much text out of the paper: instead, move some details to the appendix or the suppl. material. (take the paragraph about the slope angle threshold for example: a specialist will probably be interested in this information, but a more general audience would rather skip these details).

*We restructured the article according to the suggestions by reviewer #1. The passage on surface slope averaging was removed as we no follow another approach to determine flux directions. See answers to reviewer #1*

***Corrected** by re-structuring the article.*

Case study?

To test a new method, one should rely on the best possible data input for calibration/validation. When looking at the fields in Fig. 02 I cant really imagine that this is the case in Svalbard. It is too late for this study, but in the future I would suggest to look at more appropriate benchmark glaciers (right at our doorstep?), where data denial experiments can be realised much more easily and with much more confidence in the boundary conditions.

*We understand the reviewers concern on the input data quality on Svalbard. In ITMIX phase 2, we anticipate test glaciers for which reliable input fields will be made available in the near future. Once available, the ITMIX setup will certainly serve as a standardised benchmark for model validation.*

***No action undertaken**.*

**Specific comments**

**P2, L2** "**virtually complete coverage**" none of the cited studies (apart maybe from Paul et al, which is rather a methodological review paper) states that surface elevation change products have reached complete coverage. This is related to my general comment: the method presented in this paper is promising, but still belongs to the demanding ones in terms of data availability.

> *The specific passage, the reviewer points at, was misleading as we tried to condense to many aspects. We therefore reformulated the respective sentences giving more details and trying to be more specific.*
>
> ***Corrected*** *accordingly.*

**P2 L27** "**apparent flux divergence**" I also think that this new terminology makes no sense. One can argue about whether apparent mass-balance is the best term or not, but "apparent flux divergence" is definitely more confusing than helping.

> *Reverted terminology back to 'apparent mass balance'. For details see answers to reviewer #1.*
>
> ***Corrected****. We now use 'apparent mass balance'.*

**Figure 5** Obviously, both the observations and the ice divides (zero flux locations) are way too visible on the bedrock topography. This calls for a more constrained tuning, either by changing the way B is allowed to vary or by changing the way the model is dealing with small slopes?

> *The slope definition was changed in response to a comment from reviewer #1. This new definition will partially accommodate the slope problem. We do not see a problem with having thickness measurements imprinted in our reconstructed field. If these should be not consistent or show too high spatial variability, data pre-selection or a-priori homogenisation could be anticipated. We do not see the necessity for this here. We refrain from the suggestion to put limits on the tuning parameter B, as it accommodates for uncertainties and discrepancies in the input data as well as assumption in the approach. Therefore limits are hard to derive and they are not necessarily encountered near the divide. The slope threshold is another adjustable parameter but it is already chosen rather small as compared to other reconstruction approaches.*
>
> ***Adjusted slope computation****.*

**Figure 8** I understand the reason for using normalized values here, but from a volume estimation perspective (e.g. sea level rise), other metrics are much more important: bias and absolute error (i.e. small relative errors for large thickness values can be more relevant than large relative errors for small thickness values). Have you considered looking at absolute values, too?

> *This comment was very helpful because this figure changed often during the writing process. In its final form, it was actually again possible to present absolute values.*
>
> ***Corrected**.*

**Editorial comments**

**P3 L15** dot is missing

> ***Removed**.*

**Fig. 02** intuitively, I associate blue values with positive mass-balance / surface change and red with the opposite. Consider reversing your colortable.

> ***Corrected** by inverting colourmap.*

**All figures** consider damping the topographical shading, which is currently very strong without a clear added value.

> ***Corrected** by removing transparency from most figures. Only exception is the presentation of the second-step results.*

**All figures** consider using another colormap. Rainbow (or "jet" in python) is now considered by many as being misleading (e.g. https://www.climate-lab-book.ac.uk/2014/end-of-the-rainbow/ and many further refs online)

> ***Corrected** by changing rainbow colourbar to another colourbar provided by ColorBrewer2.0.*

---

## Referee Report (RR1)

**Referee report for 'Application of a two-step approach for mapping ice thickness to various glacier types on Svalbard'**

Douglas Brinkerhoff

June 22, 2017

In this study, Furst and others present an updated method for solving the problem of inferring ice thickness from sparse observations coupled with surface data. They then apply this method to three glacial systems on the Svalbard Archipelago. The validity of the resulting estimations of ice thickness are established with a detailed error analysis.

The paper is clearly relevant and within the appropriate scope for the journal. Contemporary interest in methods for inferring ice thickness are of great interest to the community, as evinced by numerous recent publications on the subject, including a comprehensive intercomparison (Farinotti et al., 2016). This papers contribution to the field stems primarily from its presentation of a way to circumvent some of the arduous data requirements required by the method on which it is based (Morlighem et al., 2010), and also provides an interesting real world application of the method, providing an important window into the performance of a promising algorithm with the various nuances of real observations.

In my review of the discussion paper I raised a number of points, primarily about inconsistencies in the mathematical methods and in the interpretation error. The authors have done an excellent job of correcting their methods where appropriate, or at least providing substantive discussion of the method's limitations when my requests were felt outside the scope of work. The organization of the paper has also been adapted, and the result is very clear and easy to follow. Outside of a few very minor points, I suggest publication of this manuscript.

**On temporal errors**

My comment on non-contemporaneous measurements producing an additional source of error was intended to suggest to the authors that the error bounds that they are using might generally be too small, and that they should be made larger because of this effect. My intent was not to provoke a reference to my paper, since the effect I mentioned is a pretty general principle and not a result. Instead of the reference, I'd like to see a short discussion of how (or if) non-contemporaneous measurements might increase uncertainty estimates.

**Divergence** operator notation**

 $\nabla$  is the gradient operator,  $\nabla$  is the flux divergence operator. Section 2.3.2 uses  $\nabla$ [...] as the divergence operator. Please ensure consistency and correct usage, or make the appropriate definitions.

**Typos**

**P2L14** loose  $\rightarrow$  lose **P7L30** Eq. 10 appears to be missing some things. **P9L11** to  $\rightarrow$  too

---

## Author Response (AR2)

**Response to the minor comments from A. Vieli and D. Brinkerhoff on the manuscript entitled**

**Application of a two-step approach for mapping ice thickness to various glacier types on Svalbard**

first presented on 01.03.2017

by

Fürst et al.

*First of all we want to thank A. Vieli and D. Brinkerhoff for their constructive comments. Answers are indented and in italic type while a* **short summarising reply** *is provided in italic, bold-face type.*

*Before the remaining comments of the editor and the reviewer are addressed, we want to inform the editor that the authors have* **corrected some erroneous results** *that entered the appendix. In the sensitivity analysis for the flux correction (Appendix C), the correction term should have been switched off but some dependencies were overlooked. The affected four reconstruction setups were corrected and re-run. We then updated the results in Table S1 and Fig. S5), accordingly.*

**Editor Decision: Publish subject to minor revisions**
(Editor review) (09 Jul 2017) by Andreas Vieli

**Comments to the Author:** Final decision of editor based on comments of two reviewers, major revisions undertaken and a re-review of the revised version by one of the first reviewers.

Dear Johannes Fürst,
This paper received two in depth reviews and one additional comment that were overall positive and clearly pointed out the relevance and importance and also quality of the work, but identified also some substantial issues and some substantial suggestions for revisions. Besides specific more technical points these main points concerned:

1) streamline/focus and slightly restructure manuscript

2) term apparent mass balance

3) label figures

4) error estimates (observational, formal, results discussion, etc)

5) flow direction computations

6) limitations from model physic, sliding,

7) discussion of wider use of methods/data requirements

In the revised version the authors very carefully addressed the substantial and minor technical points and explained or justified their revisions very well. Referee 1 (Brinkerhoff) also made this point in his re-review of the revised version and only has very few and mostly minor additional points which are listed. I myself also spotted some very few additional technical corrections that I listed below.

Thus the manuscript is very close to final acceptance for publication in TC but before I ask the authors to address the short list of points below.

**Re-review D. Brinkerhoff**
\*\*\*\*\*\*\*\*\*\*\*\*\*\*\*\*\*\*\*\*\*\*\*\*\*\*\*\*\*\*\*\*\*\*\*\*\*\*\*\*\*\*\*\*\*\*\*\*\*\*

In this study, Furst and others present an updated method for solving the problem of inferring ice thickness from sparse observations coupled with surface data. They then apply this method to three glacial systems on the Svalbard Archipelago. The validity of the resulting estimations of ice thickness are established with a detailed error analysis.

The paper is clearly relevant and within the appropriate scope for the journal. Contemporary interest in methods for inferring ice thickness are of great interest to the community, as evinced by numerous recent publications on the subject, including a comprehensive intercomparison (Farinotti et al., 2016). This papers contribution to the field stems primarily from its presentation of a way to circumvent some of the arduous data requirements required by the method on which it is based (Morlighem et al., 2010), and also provides an interesting real world application of the method, providing an important window into the performance of a promising algorithm with the various nuances of real observations. In my review of the discussion paper I raised a number of points, primarily about inconsistencies in the mathematical methods and in the interpretation error. The authors have done an excellent job of correcting their methods where appropriate, or at least providing substantive discussion of the method's limitations when my requests were felt outside the scope of work. The organization of the paper has also been adapted, and the result is very clear and easy to follow. Outside of a few very minor points, I suggest publication of this manuscript.

On temporal errors
My comment on non-contemporaneous measurements producing an additional source of error was in- tended to suggest to the authors that the error bounds that they are using might generally be too small, and that they should be made larger because of this effect. My intent was not to provoke a reference to my paper, since the effect I mentioned is a pretty general principle and not a result. Instead of the reference, I'd like to see a short discussion of how (or if) non-contemporaneous measurements might increase uncertainty estimates.

> ***Corrected** by adding the following brief discussion at the relevant position in the Methods section.*
>
> '*Another source of uncertainty relies in the fact that the mass conservation equation (Eq. 1) is valid only at an instant in time. Most input fields are however measured or derived over finite time intervals that naturally differ. Brinkerhoff and Aschwanden (2016) suggest that this additional uncertainty term could directly be added to the measurement error. Yet the magnitude is unclear for the individual fields and we therefore ignore it here.*'

Divergence operator notation

$\nabla$ is the gradient operator, $\nabla\cdot$ is the flux divergence operator. Section 2.3.2 uses $\nabla[\cdots]$ as the divergence operator. Please ensure consistency and correct usage, or make the appropriate definitions.

*Corrected as suggested.*

Typos

**P2 L14** loose ! lose

*Corrected as suggested.*

**P7 L30** Eq. 10 appears to be missing some things.

*Corrected '+' to '$-\nabla[F\delta n]$'.*

**P9 L11** to ! too

*Corrected as suggested.*

p 10 line 29: it should be singular: ice cap (not plural)

*Corrected as suggested.*

p. 17 line 6: should it not say: "in SMB, and surface geometry and to a flux"

*Corrected as follows:*

*'a sensitivity assessment with respect to changes in SMB and in the surface geometry as well as with respect to a flux correction'*

p. 23 line 12: Do you not refer to Fig 6c and d here? and then in the next sentence you refer to VIC only so maybe continue with: 'The ice-cap setup (VIC) shows'

*Corrected as suggested.*

Fig. 2: figure pairs ab, cd and ef all show the same unit and scale range (m/y) but to me a bit puzzling, the color-scheme used is for all different, but I can not see any reason why this should be (or is there?). Would it not be easier to compare/relate the different fields if the color scales would be the same?

*It is certainly an option to use the same colour map for all figures. Yet we refrain from it because the various fields have different physical meaning. In addition, figure pair (ef) shows the 'apparent mass balance', which is the difference between the other two figure pairs. So there is in fact no need for visual comparison. Even with the changing colour maps, the reader can identify dominant source and sink areas in the SMB and the $\partial H/\partial t$ fields.*
*Not corrected.*

P 22: Fig 6/Caption: 1) say that in a and b, black dots indicate points with used thickness measurements. And further ,I struggle a bit to distinguish between purple and black dots in c and d

*Corrected by reformulating the caption concerning the measurement colours (black, green and purple dots). In addition, a brighter shade of purple was now chosen for a better discrimination from the black colour.*

P 24 Fig 7: I would rather plot a clear (thick line) for the identity line as it is a LINE, not a area (grey area).

*Corrected as suggested.*

p. 30 line 18: all over the glacier is awkward: better say over the entire glacier area or over the whole glacier area.

*Corrected.*

p. 31 line 2, typo:  satellite (not stellite)

*Corrected as suggested.*

Appendix, p. 35, line 8:  should here not be refered to Fig. A3a (fig A1 does not show ice thickness)?

Appendix, p. 35, line 14:  similar, should here not be referred to Fig. A3b?

*Corrected both clerical errors.*

[revised manuscript text omitted]